# How Does Value Distribution in Distributional Reinforcement Learning Help Optimization?

## Abstract

We consider the problem of learning a set of probability distributions from the Bellman dynamics in distributional reinforcement learning (RL) that learns the whole return distribution compared with only its expectation in classical RL. Despite its success to obtain superior performance, we still have a poor understanding of how the value distribution in distributional RL works. In this study, we analyze the optimization benefits of distributional RL by leveraging its additional value distribution information over classical RL in the Neural Fitted Z-Iteration (Neural FZI) framework. To begin with, we demonstrate that the distribution loss of distributional RL has desirable smoothness characteristics and hence enjoys stable gradients, which is in line with its tendency to promote optimization stability. Furthermore, the acceleration effect of distributional RL is revealed by decomposing the return distribution. It turns out that distributional RL can perform favorably if the value distribution approximation is appropriate, measured by the variance of gradient estimates in each environment for any specific distributional RL algorithm. Rigorous experiments validate the stable optimization behaviors of distributional RL, contributing to its acceleration effects compared to classical RL. The findings of our research illuminate how the value distribution in distributional RL algorithms helps the optimization.

## 1 Introduction

Distributional reinforcement learning (Bellemare et al., 2017a; Dabney et al., 2018b;a; Yang et al., 2019; Zhou et al., 2020; Nguyen et al., 2020; Luo et al., 2021; Sun et al., 2022) characterizes the intrinsic randomness of returns within the framework of Reinforcement Learning (RL). When the agent interacts with the environment, the intrinsic uncertainty of the environment seeps in the the stochasticity of rewards the agent receives and the inherently chaotic state and action dynamics of physical interaction, increasing the difficulty of the RL algorithm design. Distributional RL is aimed at representing the entire distribution of returns in order to capture more intrinsic uncertainty of the environment, and therefore to use these value distributions to evaluate and optimize the policy. This is in stark contrast to the classical RL that only focuses on the expectation of the return distributions, such as temporal-difference (TD) learning (Sutton & Barto, 2018) and Q-learning (Watkins & Dayan, 1992).

As a promising branch of RL algorithms, distributional RL has demonstrated the state-of-the-art performance in a wide range of environments, e.g., Atari games, in which the representation of return distributions and the distribution divergence between the current and target return distributions within each Bellman update are pivotal to its empirical success (Dabney et al., 2018a; Sun et al., 2021b; 2022). Specifically, categorical distributional RL, e.g., C51 (Bellemare et al., 2017a; Rowland et al., 2018), integrates a categorical distribution by approximating the density probabilities in pre-specified bins with a bounded range and Kullback-Leibler (KL) divergence, serving as the first successful distributional RL family in recent years. Quantile Regression (QR) distributional RL, e.g., QR-DQN (Dabney et al., 2018b), approximates Wasserstein distance by the quantile regression loss and leverages quantiles to represent the whole return distribution. Other variants of QR-DQN, including Implicit Quantile Networks (IQN) (Dabney et al., 2018a) and Fully parameterized Quantile Function (FQF) (Yang et al., 2019), can even achieve significantly better performance across

plenty of Atari games. Moment Matching distributional RL (Nguyen et al., 2020) learns deterministic samples to evaluate the distribution distance based on Maximum Mean Discrepancy, while a more recent work called Sinkhorn distributional RL (Sun et al., 2022) interpolates Maximum Mean Discrepancy and Wasserstein distance via Sinkhorn divergence (Sinkhorn, 1967). Meanwhile, distributional RL also inherits other benefits in risk-sensitive control (Dabney et al., 2018a), policy exploration settings (Mavrin et al., 2019; Rowland et al., 2019) and robustness (Sun et al., 2021a).

Despite the remarkable empirical success of distributional RL, the illumination on its theoretical advantages is still less studied. A distributional regularization effect (Sun et al., 2021b) stemming from the additional value distribution knowledge has been characterized to explain the superiority of distributional RL over classical RL, but the benefit of the proposed regularization on the optimization of algorithms has not been investigated as the optimization plays a key role in RL algorithms. In the literature of strategies that can help the learning in RL, recent progresses mainly focus on the policy gradient methods (Sutton & Barto, 2018). Mei et al. (2020) show that the policy gradient with a softmax parameterization converges at a $\mathcal{O}(1/t)$ rate, with constants depending on the problem and initialization, which significantly expands the existing asymptotic convergence results. Entropy regularization (Haarnoja et al., 2017; 2018) has gained increasing attention as it can significantly speed up the policy optimization with a faster linear convergence rate (Mei et al., 2020). Ahmed et al. (2019) provide a fine-grained understanding on the impact of entropy on policy optimization, and emphasize that any strategy, such as entropy regularization, can only affect learning in one of two ways: either it reduces the noise in the gradient estimates or it changes the optimization landscape. These commonly-used strategies that accelerate RL learning inspire us to further investigate the optimization impact of distributional RL arising from the exploitation of return distributions.

In this paper, we study the theoretical superiority of distributional RL over classical RL from the optimization standpoint. We begin by analyzing the optimization impact of different strategies within the Neural Fitted Z-Iteration (Neural FZI) framework and point out two crucial factors that contribute to the optimization of distributional RL, including the distribution divergence and the distribution parameterization error. The smoothness property of distributional RL loss function has also been revealed leveraging the categorical parameterization, yielding its stable optimization behavior. The uniform stability in the optimization process can thus be more easily achieved for distributional RL in contrast to classical RL. In addition to the optimization stability, we also elaborate the acceleration effect of distributional RL algorithms based on the value distribution decomposition technique proposed recently. It turns out that distributional RL can be shown to speed up the convergence and perform favorably if the value distribution is approximated appropriately, which is measured by the variance of gradient estimates. Empirical results corroborate that distributional RL indeed enjoys a stable gradient behavior by observing smaller gradient norms in terms of the observations the agent encounters in the learning process. Besides, the variance reduction of gradient estimates for distributional RL algorithms with respect to network parameters also provides strong evidence to demonstrate the smoothness property and acceleration effects of distributional RL. Our study opens up many exciting research pathways in this domain through the lens of optimization, paving the way for future investigations to reveal more advantages of distributional RL.

## 2 PRELIMINARY KNOWLEDGE

**Classical RL.** In a standard RL setting, the interaction between an agent and the environment is modeled as a Markov Decision Process (MDP) $(\mathcal{S}, \mathcal{A}, R, P, \gamma)$, where $\mathcal{S}$ and $\mathcal{A}$ denote state and action spaces. $P$ is the transition kernel dynamics, $R$ is the reward measure and $\gamma \in (0, 1)$ is the discount factor. For a fixed policy $\pi$, the return, $Z^\pi = \sum_{t=0}^{\infty} \gamma^t R_t$, is a random variable representing the sum of discounted rewards observed along one trajectory of states while following the policy $\pi$. Classical RL focuses on the value function and action-value function, the expectation of returns $Z^\pi$. The action-value function $Q^\pi(s, a)$ is defined as $Q^\pi(s, a) = \mathbb{E}[Z^\pi(s, a)] = \mathbb{E}[\sum_{t=0}^{\infty} \gamma^t R(s_t, a_t)]$, where $s_0 = s, a_0 = a, s_{t+1} \sim P(\cdot|s_t, a_t)$, and $a_t \sim \pi(\cdot|s_t)$.

**Distributional RL.** Distributional RL, on the other hand, focuses on the action-value distribution, the full distribution of $Z^\pi(s, a)$ rather than only its expectation, i.e., $Q^\pi(s, a)$. Leveraging knowledge on the entire value distribution can better capture the uncertainty of returns and thus can be advantageous to explore the intrinsic uncertainty of the environment (Dabney et al., 2018a; Mavrin et al., 2019). The scalar-based classical Bellman updated is therefore extended to distributional

Bellman update, which allows a flurry of distributional RL algorithms, mainly including Categorical distributional RL (Bellemare et al., 2017a) and Quantie Regression Distributional RL (Dabney et al., 2018b;a).

**Categorical Distributional RL.**  As the first successful distributional RL family, Categorical distributional RL (Bellemare et al., 2017a) approximates the action-value distribution $\eta$ by a categorical distribution $\hat{\eta} = \sum_{i=1}^{k} f_i \delta_{l_i}$ where $l_1, l_2, ..., l_k$ is a set of fixed supports and $\{f_i\}_{i=1}^{k}$ are learnable probabilities, normally parameterized by a neural network. A projection is also introduced to have the joint support with a newly distributed target probabilities, equipped by a KL divergence to compute the distribution distance between the current and target value distribution within each Bellman update. In practice, C51 (Bellemare et al., 2017a), an instance of Categorical Distributional RL with $k = 51$, performs favorably on a wide range of environments.

**Quantile Regression (QR) Distributional RL.**  QR Distributional RL (Dabney et al., 2018b;a) approximates the value distribution $\eta$ by a mixture of Dirac $\hat{\eta} = \frac{1}{N} \sum_{i=1}^{N} \delta_{\tau_i}$, where $\tau_i = F_\eta^{-1}(\frac{2i-1}{2N})$ are the learnable quantile values at the fixed quantiles $\{\frac{2i-1}{2N}\}$ and $F^{-1}$ is the inverse cumulative distribution function of $\eta$. Since the quantile regression loss proposed in QR distributional RL can be used to approximate the Wasserstein distance, it gains favorable performance on Atari games. Moreover, the performance has been further improved by a series of variants based on quantile regression loss (Dabney et al., 2018a; Yang et al., 2019; Zhou et al., 2020). For example, Implicit Quantile Network (IQN) (Dabney et al., 2018a) utilizes a continuous mapping for the quantile function $F_\eta^{-1}(\frac{2i-1}{2N})$ rather than a fixed set of quantiles, which expands the expressiveness power of function approximators to represent the value distribution.

## 3  Optimization Effect of Distributional RL

We consider the function approximation setting to analyze the optimization benefit of distributional RL. In Section 3.1, we begin by showing the different roles of components in distributional RL on the entire optimization of RL algorithms within the Neural FZI framework. Further, in Section 3.2 we reveal the desirable smoothness properties of distributional RL loss function as opposed to classical RL, contributing to the stable optimization. Finally, the acceleration effect of distributional RL stemming from the additional value distribution is analyzed in Section 3.3, which is characterized by the variance of gradient estimates.

### 3.1  How to Optimize Neural Fitted Z-Iteration for Distributional RL?

In classical RL, *Neural Fitted Q-Iteration* (Neural FQI) (Fan et al., 2020; Riedmiller, 2005) provides a statistical interpretation of DQN (Mnih et al., 2015) while capturing its two key features, i.e., the leverage of target network and the experience replay:

$$Q_\theta^{k+1} = \operatorname*{argmin}_{Q_\theta} \frac{1}{n} \sum_{i=1}^{n} \left[ y_i - Q_\theta^k(s_i, a_i) \right]^2, \tag{1}$$

where the target $y_i = r(s_i, a_i) + \gamma \max_{a \in \mathcal{A}} Q_{\theta*}^k(s_i', a)$ is fixed within every $T_{\text{target}}$ steps to update target network $Q_{\theta*}$ by letting $\theta^* = \theta$. The experience buffer induces independent samples $\{(s_i, a_i, r_i, s_i')\}_{i \in [n]}$ and ideally without the optimization and TD approximation errors, Neural FQI is exactly the update under Bellman optimality operator (Fan et al., 2020). Similarly, in distributional RL, Sun et al. (2021b); Ma et al. (2021) proposed *Neural Fitted Z-Iteration* (Neural FZI), a distributional version of Neural FQI based on the parameterization of $Z_\theta$:

$$Z_\theta^{k+1} = \operatorname*{argmin}_{Z_\theta} \frac{1}{n} \sum_{i=1}^{n} d_p(Y_i, Z_\theta^k(s_i, a_i)), \tag{2}$$

where the target $Y_i = R(s_i, a_i) + \gamma Z_{\theta*}^k(s_i', \pi_Z(s_i'))$ is a random variable, whose distribution is also fixed within every $T_{\text{target}}$ steps. The target follows a greedy policy rule, where $\pi_Z(s_i') = \operatorname{argmax}_{a'} \mathbb{E}\left[ Z_{\theta*}^k(s_i', a') \right]$ and $d_p$ is the choice of distribution distance. Within the Neural FZI process, we can easily perceive that there are mainly two crucial components that determine the comprehensive optimization of distributional RL algorithms.

- **The choice of** $d_p$. $d_p$ in fact has two-fold impacts on the optimization of the whole Neural FZI. Firstly, $d_p$ determines the convergence rate of distributional Bellman update. For instance, distributional Bellman operator under Crámer distance is $\gamma^{\frac{1}{2}}$-contractive (Bellemare et al., 2017b), and is a $\gamma$-contraction when $d_p$ is Wasserstein distance (Bellemare et al., 2017a). Apart from the impact on the distributional Bellman update speed, $d_p$ also largely affects the continuous optimization problem to estimate parameter $\theta$ in $Z_\theta$ within each iteration of Neural FZI, including the convergence speed and the bad or good local minima issues.

- **The parameterization manner of** $Z_\theta$. The distribution representation way of $d_p$ plays an integral part of the optimization for deep RL algorithms. For example, with more expressiveness power on quantile functions, IQN outperforms QR-DQN on a wider range of Atari games, which is intuitive as a more informative representation way can approximate the true value distribution more reasonably. A smaller value distribution parameteriation error is also potential to help the optimization albeit in an indirect avenue.

Owing to the fact that convergence rates of distributional Bellman update under typical $d_p$ are basically known, our optimization analysis mainly focuses on the impact of $d_p$ and the paramterization error of $Z_\theta$ on the continuous optimization within Neural FZI of distributional RL by comparing Neural FQI of classical RL. In Sections 3.2 and 3.3, we attribute the optimization benefits of distributional RL to its distribution objective function, consisting of the aforementioned two factors, as opposed to the vanilla least squared loss in classical RL.

## 3.2 Stable Optimization Analysis based on Categorical Parameterization

To allow for a theoretical analysis, we resort to the categorical parameterization equipped with KL divergence in categorical distributional RL (Bellemare et al., 2017a), e.g., C51, in order to investigate the stable optimization properties within each iteration in Neural FZI. Concretely, we assume $Z_\theta$ is absolutely continuous and the current and target value distributions under KL divergence within a bounded range have joint supports (Arjovsky & Bottou, 2017), under which the KL divergence is well-defined. Note that this analysis strategy is slightly different from vanilla Categorical distributional RL, which also introduces a projection to redistribute probabilities of target value distribution by the neighboring smoothing without the joint support assumption. We slightly simplify Categorical distributional RL by assuming that the target distribution is still within the pre-specified support, which is still easy to satisfy in practice given a relative large bounded range $[l_0, l_k]$ in advance.

To approximate the categorical distribution, we leverage the *histogram function* $f^{s,a}$ with $k$ uniform partitions on the support to parameterize the approximated probability density function of $Z(s,a)$. With KL divergence as $d_p$, we can eventually derive the distribution objective function to be optimized within each update in Neural FZI, which is similar to the histogram distributional loss proposed in (Imani & White, 2018).

In particular, we denote $\mathbf{x}(s)$ as the state feature on each state $s$, and we let the support of $Z(s,a)$ be uniformly partitioned into $k$ bins. The output dimension of $f^{s,\cdot}$ can be $|\mathcal{A}| \times k$, where we use the index $a$ to focus on the function $f^{s,a}$. Hence, the function $f^{s,a} : \mathcal{X} \to [0,1]^k$ provides a k-dimensional vector $f^{s,a}(\mathbf{x}(s))$ of the coefficients, indicating the probability that the target is in this bin given the state feature $\mathbf{x}(s)$ and action $a$. Next, we use *softmax* based on the linear approximation $\mathbf{x}(s)^\top \theta_i$ to express $f^{s,a}$, i.e., $f_i^{s,a,\theta}(\mathbf{x}(s)) = \exp\left(\mathbf{x}(s)^\top \theta_i\right) / \sum_{j=1}^k \exp\left(\mathbf{x}(s)^\top \theta_j\right)$. For simplicity, we use $f_i^\theta(\mathbf{x}(s))$ to replace $f_i^{s,a,\theta}(\mathbf{x}(s))$. Note that the form of $f^{s,a}$ is similar to that in Softmax policy gradient optimization (Mei et al., 2020; Sutton & Barto, 2018), but here we focus on the value-based RL rather than the policy gradient RL. Our prediction probability $f_i^{s,a}$ is redefined as the probability in the $i$-th bin over the support of $Z(s,a)$, thus eventually serving as a density function. While the linear approximator is clearly limited, this is the setting where so far the cleanest results have been achieved and understanding this setting is a necessary first step towards the bigger problem of understanding distributional RL algorithms. Under this categorical parameterization equipped with KL divergence, the resulting distributional objective function $\mathcal{L}_\theta(s,a)$ for the continuous optimization for each $s,a$ pair in each iteration of Neural FZI (Eq. 2) can be expressed as:

$$\mathcal{L}_\theta(s,a) = -\sum_{i=1}^k \int_{l_i}^{l_i+w_i} p^{s,a}(y) \log \frac{f_i^\theta(\mathbf{x}(s))}{w_i} dy \propto -\sum_{i=1}^k p_i^{s,a} \log f_i^\theta(\mathbf{x}(s)), \quad (3)$$

where $\theta = \{\theta_1, ..., \theta_k\}$ and $p_i^{s,a}$ is the probability in the $i$-th bin of the true density function $p^{s,a}(x)$ for $Z(s, a)$ defined in Eq. 6. $w_i$ is the width for the $i$-th bin $(l_i, l_{i+1}]$. The derivation of the categorical distributional loss under the categorical parameterization is given in Appendix A. To attain the stable optimization property of distributional RL, we firstly derive the appealing properties of the new categorical distributional loss in Eq. 3, as shown in Proposition 1.

**Proposition 1.** *(Properties of Categorical Distributional Loss) Assume the state features $\|\mathbf{x}(s)\| \leq l$ for each state $s$, then $\mathcal{L}_\theta$ is $kl$-Lipschitz continuous, $kl^2$-smooth and convex w.r.t. the parameter $\theta$.*

Please refer to Appendix B for the proof. The derived smoothness properties of $d_p$ under the categorical distributional loss plays an integral role in the stable optimization for distributional RL. In stark contrast, classical RL optimizes a least squared loss function (Sutton & Barto, 2018) in Neural FQI. It is known that the least squared estimator has no bounded Lipschitz constant in general and is only $\lambda_{\max}$-smooth, where $\lambda_{\max}$ is the largest singular value of the design or data matrix. More specifically, for the categorical distributional loss in distributional RL, we have $\|\nabla_\theta \mathcal{L}_\theta\| \leq kl$, while the gradient norm in classical RL is $|y_i - Q_\theta^k(s, a)|\|\mathbf{x}(s)\|$, where $Q_\theta^k(s, a) = \sum_{i=1}^k (l_i + l_{i+1}) f_i^\theta(\mathbf{x}(s))/2w_i$ under the same categorical parameterization for a fair comparison. Clearly, $Q_\theta^k(s, a)$ can be sufficiently large if the support $[l_0, l_k]$ is specified to be large, which is common in environments where the agent is able to attain a high level of expected returns (Bellemare et al., 2017a). As such, $|y_i - Q_\theta^k(s, a)|$ can vary significantly more than $k$ and therefore classical RL with the potentially larger upper bound of gradient norms is prone to the instability optimization issue.

After providing the intuitive comparison in terms of gradient norms above, we next show that distributional RL loss can induce an uniform stability property under the desirable smoothness properties analyzed in Proposition 1. We recap the definition of uniform stability for an algorithm while running *Stochastic Gradient Descent* (SGD) in Definition 1.

**Definition 1.** *(Uniform Stability) (Hardt et al., 2016) Consider a loss function $g_w(z)$ parameterized by $w$ encountered on the example $z$, a randomized algorithm $\mathcal{M}$ is uniformly stable if for all data sets $\mathcal{D}, \mathcal{D}'$ such that $\mathcal{D}, \mathcal{D}'$ differ in at most one example, we have*

$$\sup_z \mathbb{E}_\mathcal{M} \left[ g_{\mathcal{M}(\mathcal{D})}(z) - g_{\mathcal{M}(\mathcal{D}')}(z) \right] \leq \epsilon_{stab}. \tag{4}$$

In Theorem 1, we show that while running SGD to solve the categorical distributional loss within each Neural FZI, the continuous optimization process in each iteration is $\epsilon_{\text{stab}}$-uniformly stable.

**Theorem 1.** *(Stable Optimization for Distributional RL) Suppose that we run SGD under $\mathcal{L}_\theta$ in Eq. 3 with step sizes $\lambda_t \leq 2/kl^2$ for $T$ steps. Assume $\|\mathbf{x}(s)\| \leq l$ for each state $s$ and action $a$, then we have $\mathcal{L}_\theta$ satisfies the uniform stability in Definition 1 with $\epsilon_{stab} \leq \frac{4kT}{n}$, i.e.,*

$$\mathbb{E} \left| \mathcal{L}_{\theta_T}(s, a) - \mathcal{L}_{\theta_T'}(s, a) \right| \leq \frac{4kT}{n}, \tag{5}$$

*where $\theta_T$ and $\theta_T'$ are the minimizers after $T$ steps under the dataset $\mathcal{D}$ and $\mathcal{D}'$, respectively.*

Please refer to the proof of Theorem 1 in Appendix C. The stable optimization has multiple advantages. In deep learning optimization literature (Hardt et al., 2016), an uniform stability can guarantee $\epsilon_{\text{stab}}$-bounded generalization gap. In reinforcement learning, algorithms with more stability tend to achieve a better final performance (Bjorck et al., 2021; Li & Pathak, 2021; Ahmed et al., 2019).

In summary, under the categorical parameterization equipped with KL divergence, the continuous optimization objective function within each update of Neural FZI for distributional RL is uniformly stable with the stability errors shrinking at the rate of $O(n^{-1})$, and the immediately obtained bounded generalization gap also guarantees a desirable local minima. This advantage can be owing to the desirable smoothness property of categorical distributional loss with a potentially smaller upper bound of gradient norms compared with classical RL. Empirically, in Section 4, we validate the stable gradient behaviors of categorical distributional RL, and similar results are also observed in Quantile Regression distributional RL. By contrast, without these smooth properties, classical RL may not yield the stable optimization property directly. For example, $\lambda_{\max}$-smooth may be of less help for the optimization given a bad conditional number of the design matrix where $\lambda_{\max}$ could be sufficiently large. The potential optimization instability for classical RL can be used to explain its inferiority to distributional RL in most environments, although it may not explain why distributional RL could not perform favorably in certain games (Ceron & Castro, 2021). We leave the comprehensive explanation as future works.

**Remark on Non-linear Categorical Parameterization.** Although the aforementioned stability optimization conclusions are established on the linear categorical parameterization on the value distribution of $Z^\pi$. Similar conclusions can be extended in the non-convex optimization case with a non-linear categorical parameterization by techniques proposed in (Hardt et al., 2016). We also empirically validate our theoretical conclusions in the experiments by directly applying practical neural network parameterized distributional RL algorithms.

## 3.3 ACCELERATION EFFECT OF DISTRIBUTIONAL RL

To characterize the acceleration effect of distributional RL, we additionally leverage the recently proposed *value distribution decomposition* (Sun et al., 2021b) to decompose the target $p^{s,a}$.

**Value Distribution Decomposition.** In order to decompose the optimization impact of value distribution into its expectation and the remaining distribution part, we adopt the wisdom from robust statistics via a variant of *gross error model* (Huber, 2004). Value distribution decomposition (Sun et al., 2021b) was successfully applied to derive the distributional regularization effect of distributional RL. We utilize $F^{s,a}$ to express the distribution function of $Z^\pi(s,a)$ and we consider the function class of $F^{s,a}$ that satisfies the following expectation decomposition:

$$F^{s,a}(x) = (1-\epsilon)\mathbb{1}_{\{x \geq \mathbb{E}[Z^\pi(s,a)]\}}(x) + \epsilon F_\mu^{s,a}(x), \tag{6}$$

where the distribution function $F_\mu^{s,a}$ is determined by $F^{s,a}$ and $\epsilon$ to measure the impact of remaining distribution *independent of* its expectation $\mathbb{E}[Z^\pi(s,a)]$. $\epsilon$ controls the proportion of $F_\mu^{s,a}(x)$ and the indicator function $\mathbb{1}_{\{x \geq \mathbb{E}[Z^\pi(s,a)]\}} = 1$ if $x \geq \mathbb{E}[Z^\pi(s,a)]$, otherwise 0. Although the function class of $F^{s,a}$ is restricted to satisfy this decomposition equality, it is still rich with the rationale rigorously demonstrated in (Sun et al., 2021b). To reveal the speeding up effect of distributional RL loss, we consider the density function form of Eq. 6, i.e., $p^{s,a}(x) = (1-\epsilon)\delta_{\{x=\mathbb{E}[Z^\pi(s,a)]\}}(x) + \epsilon\mu^{s,a}(x)$, where $\delta_{\{x=\mathbb{E}[Z^\pi(s,a)]\}}$ is a Dirac function centered at $\mathbb{E}[Z^\pi(s,a)]$ to characterize the expectation impact and $\mu^{s,a}$ is the density function of $F_\mu^{s,a}$ to measure the addition value distribution information.

Within Neural FZI, our goal is to minimize $\frac{1}{n}\sum_{i=1}^n \mathcal{L}_\theta(s_i, a_i)$. We rewrite $\mathcal{L}_\theta(s,a)$ as $\mathcal{L}_\theta(g^{s,a}, f_\theta^{s,a})$, where the target density function $g^{s,a}$ can be $p^{s,a}$, $\mu^{s,a}$ or $\delta_{\{x=\mathbb{E}[Z^\pi(s,a)]\}}$, and $f^{s,a,\theta}$ is rewritten as $f_\theta^{s,a}$ for conciseness. We denote $G^k(\theta) = \mathbb{E}\left[\mathcal{L}_\theta(\delta_{\{x=\mathbb{E}[Z^\pi(s,a)]\}}, f_\theta^{s,a})\right]$ and use $G(\theta)$ for $G^k(\theta)$ for simplicity. Based on the categorical parameterization in Section 3.2, the convex and smooth properties with respect to the parameter $\theta$ in $f_\theta$ as shown in Proposition 1 still hold for $G(\theta)$. As the KL divergence enjoys the property of unbiased gradient estimates, we let the variance of its stochastic gradient over the expectation $\delta_{\{x=\mathbb{E}[Z^\pi(s,a)]\}}$ be bounded, i.e.,

$$\mathbb{E}_{(s,a)\sim\rho^\pi}\left[\|\nabla\mathcal{L}_\theta(\delta_{\{x=\mathbb{E}[Z^\pi(s,a)]\}}, f_\theta^{s,a}) - \nabla G(\theta)\|^2\right] = \sigma^2. \tag{7}$$

Next, following the similar label smoothing analysis in (Xu et al., 2020), we further characterize the approximation degree of $f_\theta^{s,a}$ to the target value distribution $\mu^{s,a}$ by measuring its variance as $\kappa\sigma^2$:

$$\mathbb{E}_{(s,a)\sim\rho^\pi}\left[\|\nabla\mathcal{L}_\theta(\mu^{s,a}, f_\theta^{s,a}) - \nabla G(\theta)\|^2\right] = \hat{\sigma}^2 := \kappa\sigma^2. \tag{8}$$

Notably, $\kappa$ can be used to measure the approximation error between $f_\theta^{s,a}$ and $\mu^{s,a}$ and we do not assume $\hat{\sigma}^2$ to be bounded as $\kappa$ can be arbitrarily large. This expression $\kappa\sigma^2$ for $\hat{\sigma}^2$ allows us to utilize $\kappa$ to characterize different acceleration effects for distributional RL given different $\kappa$. Concretely, a favorable approximation of $f_\theta^{s,a}$ to $\mu^{s,a}$ would lead to a small $\kappa$ that contributes to the acceleration effect of distributional RL as shown in Theorem 2.

**Proposition 2.** *Based on the value distribution decomposition in Eq. 6, and Eq. 8, we have:*

$$\mathbb{E}_{(s,a)\sim\rho^\pi}\left[\|\nabla\mathcal{L}_\theta(p^{s,a}, f_\theta^{s,a}) - \nabla G(\theta)\|^2\right] \leq (1-\epsilon)^2\sigma^2 + \epsilon^2\kappa\sigma^2. \tag{9}$$

Based on Eq. 8, we immediately have Proposition 2 with proof in Appendix D for the proof. Before comparing the sample complexity in the optimization process of both classical and distributional RL, we provide the definition of the first-order $\tau$-stationary point, which is preferred in the optimization of deep learning rather than the a simple stationary point in order to guarantee the generalization.

**Definition 2.** *(First-order $\tau$-Stationary Point) While solving $\min_\theta G(\theta)$, the updated parameters $\theta_T$ after $T$ steps is a first-order $\tau$-stationary point if $\|\nabla G(\theta_T)\| \leq \tau$, where the small $\tau$ is in $(0,1)$.*

Based on Definition 2, we formally characterize the acceleration effects for distributional RL in Theorem 2 that depends upon approximation errors between $\mu^{s,a}$ and $f_\theta^{s,a}$ measured by $\kappa$.

**Theorem 2.** *(Sample Complexity and Acceleration Effects of Distributional RL) While running SGD to minimize $\mathcal{L}_\theta$ in Eq. 6 within Neural FZI, we assume the step size $\lambda = 1/kl^2$, $\epsilon = 1/(1+\kappa)$ across (2) and (3), and the sample is uniformly drawn from $T$ samples, then:*

*(1) (**Classical RL**) When minimizing $\mathcal{L}_\theta(\delta_{\{x=\mathbb{E}[Z^\pi(s,a)]\}}, f_\theta^{s,a})$, $T = O(\frac{1}{\tau^4})$ such that $\mathcal{L}_\theta$ converges to a $\tau$-stationary point in expectation.*

*(2) (**Distributional RL with $\kappa \leq \frac{\tau}{2\sigma}$**) When minimizing $\mathcal{L}_\theta(p^{s,a}, f_\theta^{s,a})$, let $T = \frac{4G(\theta_0)}{\lambda \tau^2} = O(\frac{1}{\tau^2})$, $\mathcal{L}_\theta$ converges to a $\tau$-stationary point in expectation.*

*(3) (**Distributional RL with $\kappa > \frac{\tau}{2\sigma}$**) When minimizing $\mathcal{L}_\theta(p^{s,a}, f_\theta^{s,a})$, let $T = \frac{G(\theta_0)}{\lambda \kappa^2 \sigma^2} = O(\frac{1}{\tau^2})$, $\mathcal{L}_\theta$ does not converge to a $\tau$-stationary point, but can guarantee a $O(\kappa^2)$-stationary point.*

The proof is provided in Appendix E. Theorem 2 is inspired by the intuitive connection between the value distribution in distributional RL and the label distribution in label smoothing technique (Xu et al., 2020). Importantly, Theorem 2 demonstrates that solving categorical distributional loss of distributional RL can speed up the convergence if a distribution approximation error is favorable. Otherwise, the convergence point, albeit stationary, may not guarantee a desirable performance under an agnostic $\kappa$, which may be very large on certain environments.

**In Classical RL scenario**, we provide an equivalence between $\mathcal{L}_\theta(\delta_{\{x=\mathbb{E}[Z^\pi(s,a)]\}}, f_\theta^{s,a})$ and mean squared error loss (Eq. 1) in Neural FQI in Appendix H. **In the first scenario ((2) in Theorem 2)**, there is only a small approximation or paramterization error between $f_\theta^{s,a}$ and $p^{s,a}$ (or $\mu^{s,a}$), corresponding to a small $\kappa$ with $\kappa \leq \frac{\tau}{2\sigma}$. In this case, solving $\mathcal{L}_\theta$ based on the categorical parameterization can reduce the sample complexity from $O(\frac{1}{\tau^4})$ to $O(\frac{1}{\tau^2})$ compared with classical RL in (1) of Theorem 2, and meanwhile guarantees a $\tau$-stationary point. **In the second scenario ((3) in Theorem 2)** especially for some challenging environments with much intrinsic uncertainty, we can also attain a relatively large approximation error or parameterization error of $Z_\theta$ with a large $\kappa > \frac{\tau}{2\sigma}$ as the distributional TD approximation error could be potentially large in practice. Under this circumstance, distributional RL algorithms may fail to speed up the convergence or achieve the superior performance compared with classical RL as $\mathcal{O}(\kappa^2)$ could be potentially large on some complex environments. If $\mathcal{O}(\kappa^2)$ is proper, distributional RL can still potentially perform reasonably due the to $\mathcal{O}(\kappa^2)$-stationary point guarantee.

Theses theoretical results also coincide with past empirical observations (Dabney et al., 2018b; Ceron & Castro, 2021), where distributional RL algorithms outperform classical RL in most cases, but are inferior in certain environments. Based on our results in Theorem 2, we contend that these certain environments have much intrinsic uncertainty, the distribution parameterization error between $Z_\theta$ and the true value distribution under the distributional TD approximation is still too large ($\kappa > \frac{\tau}{2\sigma}$) to guarantee a favorable convergence point for distributional RL algorithms with different $d_p$, which is intuitive.

## 4 EXPERIMENTS

We perform extensive experiments on eight continuous control MuJoCo games to validate the theoretical optimization advantage of distributional RL algorithms analyzed in Section 3, including the stable gradient behaviors of distributional RL to achieve the uniform stability as well as the acceleration effects determined by the distribution parameterization error.

**Implementation.** Our implementation is based Soft Actor Critic (SAC) (Haarnoja et al., 2018) and distributional Soft Actor Critic (Ma et al., 2020). We eliminate the optimization impact of entropy regularization in these algorithm implementations, and thus we denote the resulting algorithms as Actor Critic (AC) and Distributional Actor Critic (DAC) for the conciseness. For DAC, we firstly perform the C51 algorithm to the critic to extend the classical critic loss to the distributional version denoted by *DAC (C51)* as our theoretical analysis in Sections 3.2 and 3.3 are mainly based on categorical parameterization. We further apply our empirical demonstration on Quantile Regression distributional RL heuristically, i.e., Implicit Quantile Network (IQN), which is denoted as *DAC (IQN)*. Hyper-parameters and more implementation details are provided in Appendix F.

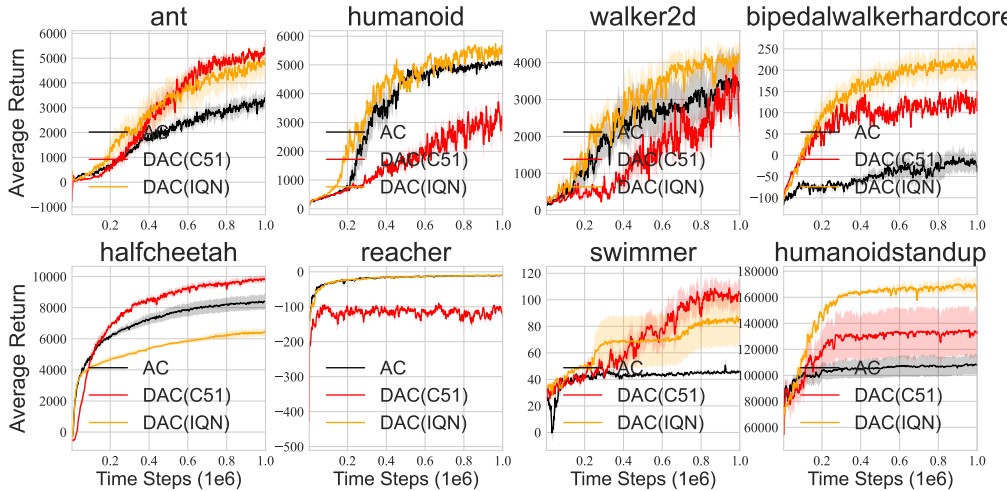

Figure 1: **Performance.** Learning curve of AC, DAC (C51) and DAC (IQN) over 5 seeds with smooth size 5 across eight MuJoCo games.

## 4.1 PERFORMANCE AND UNIFORM STABILITY IN DISTRIBUTIONAL RL OPTIMIZATION

Figure 1 suggests that DAC (IQN) in orange lines outperforms its classical version AC (black lines) across all environments, while DAC (C51) in red lines is inferior to AC on humanoid, walker2d and reacher. This could be explained by a more flexible parameterization of IQN over C51.

We then demonstrate the advantage of uniform optimization stability for distributional RL over classical RL. According to Theorem 1, the stable optimization of distribution loss with Neural FZI is described as a bounded loss difference for a neighboring dataset in terms of each state $s$ and action $a$. In other words, the error bound holds by taking the supreme over each state the agent encounters. To measure this algorithm stability, while far from perfect, we consider to leverage *the average gradient norms with respect to the state feature* $\mathbf{x}(s)$ in the whole optimization process as the proxy due to the fact that the gradient could measure the sensitivity of loss function regarding each state the agent observes. From Figure 2, it turns out that both DAC (C51) and DAC (IQN) entail a much smaller gradient norm magnitude as opposed to their classical version AC (black

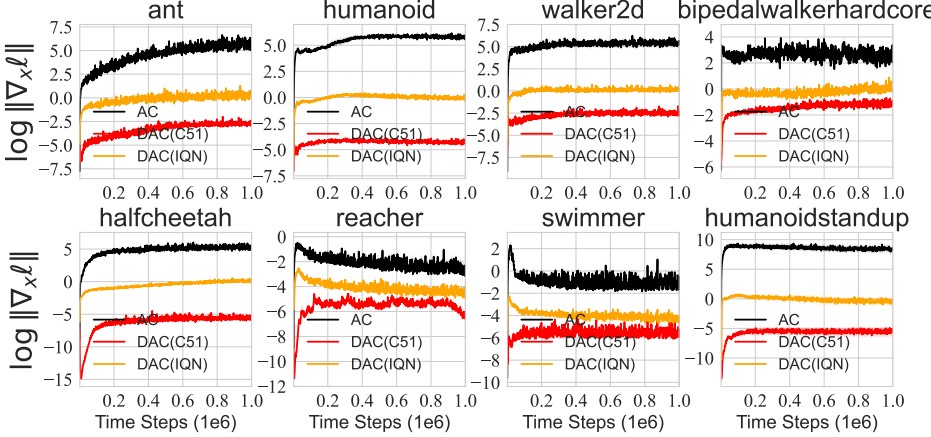

Figure 2: **Stable Optimization.** The critic gradient norms in the logarithmic scale regarding *the state* during the training of AC, DAC (C51), DAC (IQN) over 5 seeds on eight MuJoCo environments.

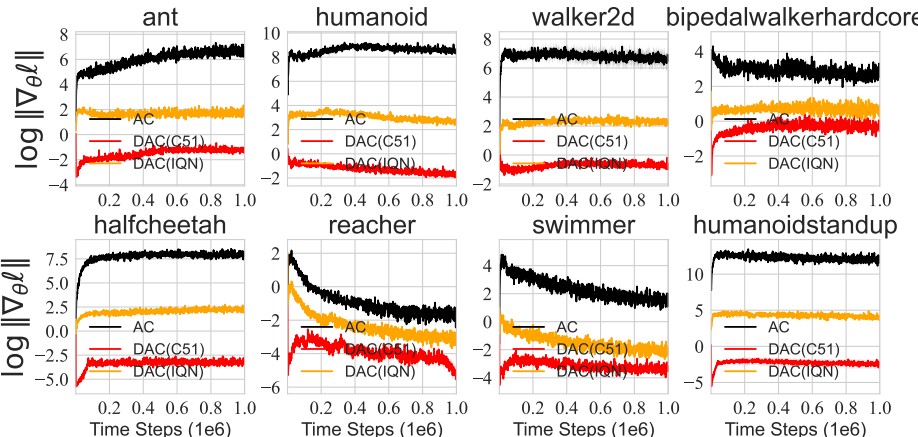

Figure 3: **Acceleration Effect.** The critic gradient norms in the logarithmic scale regarding *network parameters* in the training of AC, DAC (C51), DAC (IQN) over 5 seeds on MuJoCo environments.

lines) across all eight MuJoCo environments, which corroborates the theoretical advantage of the uniform optimization stability for distributional RL analyzed in Theorem 1.

## 4.2 SMOOTHNESS PROPERTY AND ACCELERATION EFFECT OF DISTRIBUTIONAL RL

Theorem 2 demonstrates that distributional RL can speed up the convergence if the distribution parameterization is appropriate, characterized by the variance of the gradient estimates with a small $\kappa$ (case (2) in Theorem 2). To demonstrate it, we use the proxy by evaluating the $\ell_2$-norms of gradients *with respect to network parameters* of the critic for AC and DAC. We mainly focus on a direct comparison between vanilla AC and DAC algorithm, although their network architectures are slightly different. Similar results under the same architecture and via the value distribution decomposition of Eq. 6 are provided in Appendix G.

Figure 3 showcases that both DAC (C51) and DAC (IQN) have smaller gradient norms in terms of network parameters $\theta$ compared with AC in the whole optimization process, which directly validates that distributional RL loss is more likely to enjoy smoothness properties in Proposition 1. In terms of acceleration effects, the property of stationary points, albeit being different, in cases (2) and (3) of Theorem 2 guarantees bounded gradient norms, but the precise evaluation of $\kappa$ is tricky in order to discriminate either case (2) or (3) for each algorithm in a specific environment. Nevertheless, by considering the fact that DAC (IQN) outperforms DAC (C51) in most environments in Figure 1, we hypothesize that the inferiority of DAC (C51) on humanoid, walker2d and reacher could be owing to its larger parameterization errors $\kappa$ in these environments. This results in the worse performance of DAC (C51) compared with DAC (IQN) that is more likely to accord with the case (3) in Theorem 2 due to its richer distribution expressiveness power than C51.

## 5 DISCUSSIONS AND CONCLUSION

Our optimization analysis of distributional RL is based on the categorical parameterization, and the alternative analysis on Wasserstein distance can be an integral complementary for our conclusions. Acceleration effects could be further investigated to explain whether a typical distributional RL algorithm can perform favorably in a specific environment. We leave them as future works.

In our paper, we answer the question: *how does value distribution in distributional RL help the optimization* from two perspectives, including the stable optimization analysis based on the smoothness property of categorical distributional loss, as well as the acceleration effects determined by the variance of gradient estimates. We theoretically and empirically show that distributional RL embraces stable gradient behaviors and could speed up the convergence if the distribution approximation is desirable or the parameterization error is sufficiently small.

**Ethics Statement.** Due to the fact that our study is about the theoretical properties of distributional RL algorithms, we do not think our research is involved with any ethics issues.

**Reproducibility Statement.** As stated in Section 4, our implementation is based on the public code of SAC (Haarnoja et al., 2018) and Distributional SAC (Ma et al., 2020). We also provide implementation details in Appendix F for reproducibility. For the theoretical results, rigorous proof is also given in Appendix from A to E.

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

## A    DERIVATION OF CATEGORICAL DISTRIBUTIONAL LOSS

We show the derivation details of the Categorical distribution loss starting from KL divergence between $p$ and $q_\theta$. $p_i$ is the cumulative probability increment of target distribution $\{Y_i\}_{i\in[n]}$ within the $i$-th bin, and $q_\theta$ corresponds to a (normalized) histogram, and has density values $\frac{f_i^\theta(\mathbf{x}(s))}{w_i}$ per bin. Thus, we have:

$$
\begin{aligned}
D_{\mathrm{KL}}\left(p^{s,a}, q_\theta^{s,a}\right) &= \int_a^b p^{s,a}(y)\log p^{s,a}(y)dy - \int_a^b p^{s,a}(y)\log q_\theta^{s,a}(y)dy \\
&\propto -\int_a^b p^{s,a}(y)\log q_\theta^{s,a}(y)dy \\
&= -\sum_{i=1}^k \int_{l_i}^{l_i+w_i} p^{s,a}(y)\log \frac{f_i^\theta(\mathbf{x}(s))}{w_i}dy \\
&= -\sum_{i=1}^k \log \frac{f_i^\theta(\mathbf{x}(s))}{w_i}\underbrace{\left(F^{s,a}\left(l_i+w_i\right) - F^{s,a}\left(l_i\right)\right)}_{p_i^{s,a}} \\
&\propto -\sum_{i=1}^k p_i^{s,a}\log f_i^\theta(\mathbf{x}(s))
\end{aligned}
\tag{10}
$$

where the first $\propto$ results from the fixed target $p^{s,a}$ in the Neural FZI framework. The second equality is based on the categorical parameterization for the density function $q_\theta^{s,a}$. The last $\propto$ holds because the width parameter $w_i$ can be ignored for this minimization problem.

## B    PROOF OF PROPOSITION 1

*Proof.* For the Categorical distributional loss below,

$$
\mathcal{L}_\theta(s,a) = -\sum_{i=1}^k p_i^{s,a}\log f_i^\theta(\mathbf{x}(s)), \text{ where } f_i^\theta(\mathbf{x}(s)) = \frac{\exp\left(\mathbf{x}(s)^\top \theta_i\right)}{\sum_{j=1}^k \exp\left(\mathbf{x}(s)^\top \theta_j\right)}
$$

**(1) Convexity.** Note that $-\log\frac{\exp(\mathbf{x}(s)^\top \theta_i)}{\sum_{j=1}^k \exp(\mathbf{x}(s)^\top \theta_j)} = \log\sum_{j=1}^k \exp\left(\mathbf{x}(s)^\top \theta_j\right) - \mathbf{x}(s)^\top \theta_i$, the first term is Log-sum-exp, which is convex (see Convex optimization by Boyd and Vandenberghe), and the second term is affine function. Thus, $\mathcal{L}_\theta(s,a)$ is convex.

**(2) $\mathcal{L}_\theta(s,a)$ is $kl$-Lipschitz continuous.** We compute the gradient of the Histogram distributional loss regarding $\theta_i$:

$$
\begin{aligned}
&\frac{\partial}{\partial \theta_i}\sum_{j=1}^k p_j^{s,a}\log f_j^\theta(\mathbf{x}(s)) \\
&= \sum_{j=1}^k p_j^{s,a}\frac{1}{f_j^\theta(\mathbf{x}(s))}\nabla_{\theta_i}f_j^\theta(\mathbf{x}(s)) \\
&= \sum_{j=1}^k p_j^{s,a}\frac{1}{f_j^\theta(\mathbf{x}(s))}f_i^\theta(\mathbf{x}(s))(\delta_{ij} - f_j^\theta(\mathbf{x}(s)))\mathbf{x}(s) \\
&= \left(p_i^{s,a}(1 - f_i^\theta(\mathbf{x}(s))) - \sum_{j\neq i}^k p_j^{s,a}f_i^\theta(\mathbf{x}(s))\right)\mathbf{x}(s) \\
&= \left(p_i^{s,a} - p_i^{s,a}f_i^\theta(\mathbf{x}(s)) - (1 - p_i^{s,a})f_i^\theta(\mathbf{x}(s))\right)\mathbf{x}(s) \\
&= \left(p_i^{s,a} - f_i^\theta(\mathbf{x}(s))\right)\mathbf{x}(s)
\end{aligned}
\tag{11}
$$

where $\delta_{ij} = 1$ if $i = j$, otherwise 0. Then, as we have $\|\mathbf{x}(s)\| \le l$, we bound the norm of its gradient

$$
\begin{aligned}
&\|\frac{\partial}{\partial \theta} \sum_{j=1}^{k} p_j \log f_j^{\theta}(\mathbf{x}(s))\| \\
&\le \sum_{i=1}^{k} \|\frac{\partial}{\partial \theta_i} \sum_{j=1}^{k} p_j \log f_j^{\theta}(\mathbf{x}(s))\| \\
&= \sum_{i=1}^{k} \| \left( p_i^{s,a} - f_i^{\theta}(\mathbf{x}(s)) \right) \mathbf{x}(s)\| \\
&\le \sum_{i=1}^{k} |p_i^{s,a} - f_i^{\theta}(\mathbf{x}(s))| \|\mathbf{x}(s)\| \\
&\le kl
\end{aligned}
\tag{12}
$$

The last equality satisfies because $|p_i - f_i^{\theta}(\mathbf{x}(s))|$ is less than 1 and even smaller. Therefore, we obtain that $\mathcal{L}_{\theta}$ is $kl$-Lipschitz.

**(3) $\mathcal{L}_{\theta}$ is $kl^2$-Lipschitz smooth.** A lemma is that $\log(1 + \exp(x))$ is $\frac{1}{4}$-smooth as its second-order gradient is bounded by $\frac{1}{4}$, and if $g(w)$ is $\beta$-smooth w.r.t. $w$, then $g(\langle x, w \rangle)$ is $\beta\|x\|^2$-smooth. Based on this knowledge, we firstly focus on the 1-dimensional case of function $\log f_j^{\theta}(z)$, where $f_j^{\theta}(z) = \frac{\exp z_j}{\sum_{i=1}^{k} \exp z_i}$. As we have derived, we know that $\frac{\partial}{\partial \theta_i} \log f_j^{\theta}(z_j) = \delta_{ij} - f_i^{\theta}(z_i)$. Then the second-order gradient is $\frac{\partial^2 \log f_j^{\theta}(z)}{\partial \theta_i \partial \theta_k} = -f_i^{\theta}(z)(\delta_{ik} - f_k^{\theta}(z)) = f_i^{\theta}(z)(f_k^{\theta}(z) - 1)$ if $i = k$, otherwise $f_i^{\theta}(z)f_k^{\theta}(z)$. Clearly, $|\frac{\partial^2 \log f_j^{\theta}(z)}{\partial \theta_i \partial \theta_k}| \le 1$, which implies that $\log f_j^{\theta}(z)$ is 1-smooth. Thus, $\log f_j^{\theta}(\langle x, \theta_i \rangle)$ is $\|x\|^2$-smooth, or $l^2$-smooth. Further, $\sum_{j=1}^{k} p_j^{s,a} \log f_j^{\theta}(\mathbf{x}(s))$ is also $l^2$-smooth as we have

$$
\begin{aligned}
&\|\nabla_{\theta_i} \sum_{j=1}^{k} p_j^{s,a} \log f_j^{\mu}(\mathbf{x}(s)) - \nabla_{\theta_i} \sum_{j=1}^{k} p_j^{s,a} \log f_j^{\nu}(\mathbf{x}(s))\| \\
&\le \sum_{j=1}^{k} p_j^{s,a} \|\nabla_{\theta_i} \log f_j^{\mu}(\mathbf{x}(s)) - \nabla_{\theta_i} \log f_j^{\nu}(\mathbf{x}(s))\| \\
&\le \sum_{j=1}^{k} p_j^{s,a} \cdot l^2 \|\mu - \nu\| \\
&= l^2 \|\mu - \nu\|
\end{aligned}
\tag{13}
$$

for each parameter $\mu$ and $\nu$. Therefore, we further have

$$
\begin{aligned}
&\|\nabla_{\theta} \sum_{j=1}^{k} p_j^{s,a} \log f_j^{\mu}(\mathbf{x}(s)) - \nabla_{\theta} \sum_{j=1}^{k} p_j^{s,a} \log f_j^{\nu}(\mathbf{x}(s))\| \\
&\le \sum_{i=1}^{k} \|\nabla_{\theta_i} \sum_{j=1}^{k} p_j^{s,a} \log f_j^{\mu}(\mathbf{x}(s)) - \nabla_{\theta_i} \sum_{j=1}^{k} p_j^{s,a} \log f_j^{\nu}(\mathbf{x}(s))\| \\
&\le \sum_{i=1}^{k} l^2 \|\mu - \nu\| \\
&= kl^2 \|\mu - \nu\|
\end{aligned}
\tag{14}
$$

Finally, we conclude that $\mathcal{L}_{\theta}(s, a)$ is $kl^2$-smooth.

$\square$

## C    PROOF OF THEOREM 1

*Proof.* Consider the stochastic gradient descent rule as $G_{\lambda,\mathcal{L}}(\theta) = \theta - \lambda\nabla_\theta\mathcal{L}_\theta$. Firstly, we provide two definitions about $\mathcal{L}_\theta$ for the following proof.

**Definition 3.** *($\sigma$-bounded) An update rule is $\sigma$-bounded if $\sup_\theta \|\theta - \lambda\nabla_\theta\mathcal{L}_\theta\| \leq \sigma$.*

**Definition 4.** *($\eta$-expansive) An update rule is $\eta$-expansive if $\sup_{v,w} \frac{\|G_{\lambda,\mathcal{L}}(v)-G_{\lambda,\mathcal{L}}(w)\|}{\|u-w\|} \leq \eta$.*

**Lemma 1.** *(Grow Recursion, Lemma 2.5 (Hardt et al., 2016)) Fix an arbitrary sequence of updates $G_1, ..., G_T$ and another sequence $G'_1, ..., G'_T$. Let $\theta_0 = \theta'_0$ be the starting point and define $\delta_t = \|\theta'_i - \theta_t\|$, where $\theta_t$ and $\theta'_t$ are defined recursively through*

$$\theta_{t+1} = G_{\lambda,\mathcal{L}}(\theta_t), \ \theta'_{t+1} = G'_{\lambda,\mathcal{L}}(\theta'_t)$$

*Then we have the recurrence relation:*

$$\delta_{t+1} \leq \begin{cases} \eta\delta_t & G_t = G'_t \text{ is } \eta\text{-expansive} \\ \min(\eta, 1)\delta_t + 2\sigma_t & G_t \text{ and } G'_t \text{ are } \sigma\text{-bounded}, G_t \text{ is } \eta \text{ expansive} \end{cases}$$

**Lemma 2.** *(Lipschitz Continuity) Assume $\mathcal{L}_\theta$ is $L$-Lipschitz, the gradient update $G_{\lambda,\mathcal{L}}$ is $(\lambda L)$-bounded.*

*Proof.* $\|\theta - G_{\lambda,\mathcal{L}}(\theta)\| = \|\lambda\nabla_\theta\mathcal{L}_\theta\| \leq \lambda L$    □

**Lemma 3.** *(Lipschitz Smoothness) Assume $\mathcal{L}_\theta$ is $\beta$-smooth, then for any $\lambda \leq \frac{2}{\beta}$, the gradient update $G_{\lambda,\mathcal{L}}$ is 1-expansive.*

*Proof.* Please refer to Lemma 3.7 in (Hardt et al., 2016) for the proof.    □

Based on all the results above, we start to prove Theorem 1. Our proof is largely based on (Hardt et al., 2016), but it is applicable in distributional RL setting as well as considering desirable properties of histogram distributional loss. According to Proposition 1, we attain that $\mathcal{L}_\theta$ is $kl$-Lipschitz as well as $kl^2$-smooth, and thus based on Lemma 2 and Lemma 3, we have $G_{\lambda,\mathcal{L}}$ is $(\lambda kl)$-bounded, and 1-expansive if $\lambda \leq \frac{2}{kl^2}$. In the step $t$, SGD selects samples that are both in $\mathcal{D}$ and $\mathcal{D}'$, with probability $1 - \frac{1}{n}$. In this case, $G_t = G'_t$, and thus $\delta_{t+1} \leq \delta_t$ as $G_t$ is 1-expansive based on Lemma 1. The other case is that samples selected are different with probability $\frac{1}{n}$, where $\delta_{t+1} \leq \delta_t + 2\lambda_t kl$ based on Lemma 1. Thus, if $\lambda_t \leq \frac{2}{kl^2}$, for each state $s$ and action $a$, we have:

$$\mathbb{E}\left|\mathcal{L}_{\theta_T}(s,a) - \mathcal{L}_{\theta'_T}(s,a)\right| \leq kl\mathbb{E}\left[\delta_T\right], \text{ where } \delta_T = \|\theta_T - \theta'_T\|$$

$$\leq kl\left((1 - \frac{1}{n})\mathbb{E}\left[\delta_{T-1}\right] + \frac{1}{n}\mathbb{E}\left[\delta_{T-1}\right] + \frac{2\lambda_{T-1}kl}{n}\right)$$

$$= kl\left(\mathbb{E}\left[\delta_{T-1}\right] + \frac{2\lambda_{T-1}kl}{n}\right)$$

$$= kl\left(\mathbb{E}\left[\delta_0\right] + \sum_{t=0}^{T-1}\frac{2\lambda_t kl}{n}\right) \tag{15}$$

$$\leq \frac{2k^2l^2}{n}\sum_{t=0}^{T-1}\frac{2}{kl^2}$$

$$= \frac{4kT}{n}$$

Since this bounds hold for all $\mathcal{D}$, $\mathcal{D}'$ and $s, a$, we attain the uniform stability in Definition 1 for our categorical distributional loss applied in distributional RL.

Define the population risk as:

$$R[\theta] = \mathbb{E}_x\mathcal{L}_\theta(s,a)$$

and the empirical risk as:

$$R_S[\theta] = \frac{1}{n} \sum_{i=1}^{n} \mathcal{L}_\theta(s_i, a_i)$$

According to Theorem 2.2 in (Hardt et al., 2016), if an algorithm $\mathcal{M}$ is $\epsilon_{\text{stab}}$-uniformly stable, then the generalization gap is $\epsilon_{\text{stab}}$-bounded, i.e.,

$$|\mathbb{E}_{S,A}[R_S[\mathcal{M}(\mathcal{D})] - R[\mathcal{M}(\mathcal{D}')]]| \le \epsilon_{\text{stab}}$$

$\square$

## D   PROOF OF PROPOSITION 2

$$\mathbb{E}_{(s,a)\sim\rho^\pi}\left[\|\nabla\mathcal{L}_\theta(p^{s,a}, f_\theta^{s,a})) - \nabla G(\theta)\|^2\right] \le (1-\epsilon)^2\sigma^2 + \epsilon^2\kappa\sigma^2. \tag{16}$$

*Proof.* As we know that $p^{s,a}(x) = (1-\epsilon)\delta_{\{x=\mathbb{E}[Z^\pi(s,a)]\}}(x) + \epsilon\mu^{s,a}(x)$ and we use KL divergence in $\mathcal{L}_\theta$, then we have:

$$\nabla\mathcal{L}_\theta(p^{s,a}, f_\theta^{s,a}) = (1-\epsilon)\nabla\mathcal{L}_\theta(\delta_{\{x=\mathbb{E}[Z^\pi(s,a)]\}}, f_\theta^{s,a}) + \epsilon\nabla\mathcal{L}_\theta(\mu^{s,a}, f_\theta^{s,a})$$

Therefore,

$$\mathbb{E}_{(s,a)\sim\rho^\pi}\left[\|\nabla\mathcal{L}_\theta(p^{s,a}, f_\theta^{s,a}))- \nabla G(\theta)\|^2\right]$$
$$\le \mathbb{E}_{(s,a)\sim\rho^\pi}\left[(1-\epsilon)^2\|\nabla\mathcal{L}_\theta(\delta_{\{x=\mathbb{E}[Z^\pi(s,a)]\}}, f_\theta^{s,a})) - \nabla G(\theta)\|^2 + \epsilon^2\|\nabla\mathcal{L}_\theta(\mu^{s,a}, f_\theta^{s,a})) - \nabla G(\theta)\|^2\right]$$
$$= (1-\epsilon)^2\sigma^2 + \epsilon^2\kappa\sigma^2,$$
$$\tag{17}$$

where the first inequality uses the triangle inequality of norm, i.e., $\|(1-\epsilon)\mathbf{a}+\epsilon\mathbf{b}\|^2 \le (1-\epsilon)^2\|\mathbf{a}\|^2 + \epsilon^2\|\mathbf{b}\|^2$, and the last equality uses the definition of the variance of $\mathcal{L}_\theta(\delta_{\{x=\mathbb{E}[Z^\pi(s,a)]\}}, f_\theta^{s,a})$ and $\mathcal{L}_\theta(\mu^{s,a}, f_\theta^{s,a})$. $\square$

## E   PROOF OF THEOREM 2

*Proof.* (1) If we only consider the expectation of $Z^\pi(s,a)$, we use the information $\delta_{\{x=\mathbb{E}[Z^\pi(s,a)]\}}$ to construct the loss function. As $\mathcal{L}_\theta(\delta_{\{x=\mathbb{E}[Z^\pi(s,a)]\}}, q_\theta^{s,a})$ is $kl^2$-smooth, we have

$$G(\theta_{t+1}) - G(\theta_t)$$
$$\le \langle\nabla G(\theta_t), \theta_{t+1} - \theta_t\rangle + \frac{kl^2}{2}\|\theta_{t+1} - \theta_t\|^2 \tag{18}$$
$$= -\lambda\langle\nabla G(\theta_t), \nabla\mathcal{L}_\theta(\delta_{\{x=\mathbb{E}[Z^\pi(s,a)]\}}, f_\theta^{s,a})\rangle + \frac{kl^2\lambda^2}{2}\|\nabla\mathcal{L}_\theta(\delta_{\{x=\mathbb{E}[Z^\pi(s,a)]\}}, f_\theta^{s,a})\|^2$$

where the last first equation is according to the definition of Lipschitz-smoothness, and the last second one is based on the updating rule of $\theta$. Next, we take the expectation on both sides,

$$\mathbb{E}[G(\theta_{t+1}) - G(\theta_t)]$$
$$\le -\lambda\mathbb{E}\left[\|\nabla G(\theta_t)\|^2\right] + \frac{kl^2\lambda^2}{2}\mathbb{E}\left[\|\nabla\mathcal{L}_\theta(\delta_{\{x=\mathbb{E}[Z^\pi(s,a)]\}}, f_\theta^{s,a}) - \nabla G(\theta_t) + \nabla G(\theta_t)\|^2\right]$$
$$\le -\lambda\mathbb{E}\left[\|\nabla G(\theta_t)\|^2\right] + \frac{kl^2\lambda^2}{2}\mathbb{E}\left[\|\nabla\mathcal{L}_\theta(\delta_{\{x=\mathbb{E}[Z^\pi(s,a)]\}}, f_\theta^{s,a}) - \nabla G(\theta_t)\|^2\right] + \frac{kl^2\lambda^2}{2}\mathbb{E}\left[\|\nabla G(\theta_t)\|^2\right]$$
$$= \frac{\lambda(kl^2\lambda - 2)}{2}\mathbb{E}\left[\|\nabla G(\theta_t)\|^2\right] + \frac{kl^2\lambda^2}{2}\sigma^2$$
$$\le -\frac{\lambda}{2}\mathbb{E}\left[\|\nabla G(\theta_t)\|^2\right] + \frac{kl^2\lambda^2}{2}\sigma^2$$
$$\tag{19}$$

where the first two equation hold because $\nabla G(\theta) = \mathbb{E}[\nabla\mathcal{L}_\theta]$ and the last inequality comes from $\lambda \le \frac{1}{kl^2}$. Through the summation, we obtain that

$$\mathbb{E}[G(\theta_T) - G(\theta_0)] \le -\frac{\lambda}{2}\sum_{t=0}^{T-1}\mathbb{E}\left[\|\nabla G(\theta_t)\|^2\right] + \frac{kl^2\lambda^2 T}{2}\sigma^2$$

We let $\mathbb{E}\left[G(\theta_T)\right] = 0$, we have

$$\frac{1}{T}\sum_{t=0}^{T-1}\mathbb{E}\left[\|\nabla G(\theta_t)\|^2\right] \leq \frac{2G(\theta_0)}{\lambda T} + kl^2\lambda\sigma^2$$

By setting $\lambda \leq \frac{\tau^2}{2kl^2\sigma^2}$ and $T = \frac{4G(\theta_0)}{\lambda\tau^2}$, we can have $\frac{1}{T}\sum_{t=0}^{T-1}\mathbb{E}\left[\|\nabla G(\theta_t)\|^2\right] \leq \tau^2$, implying that the degenerated loss function based on the expectation $\delta_{\{x=\mathbb{E}[Z^\pi(s,a)]\}}$ can achieve $\tau$-stationary point if the sample complexity $T = O(\frac{1}{\tau^4})$.

(2) and (3) We are still based on the $kl^2$-smoothness of $\mathcal{L}(p^{s,a}, f_\theta^{s,a})$.

$$G(\theta_{t+1}) - G(\theta_t)$$
$$\leq \langle \nabla G(\theta_t), \theta_{t+1} - \theta_t \rangle + \frac{kl^2}{2}\|\theta_{t+1} - \theta_t\|^2$$
$$= -\lambda\langle\nabla G(\theta_t), \nabla\mathcal{L}_\theta(p^{s,a}, f_\theta^{s,a})\rangle + \frac{kl^2\lambda^2}{2}\|\nabla\mathcal{L}_\theta(p^{s,a}, f_\theta^{s,a})\|^2 \qquad (20)$$
$$= -\frac{\lambda}{2}\|\nabla G(\theta_t)\|^2 + \frac{\lambda}{2}\|\nabla G(\theta_t) - \nabla\mathcal{L}_\theta(p^{s,a}, f_\theta^{s,a})\|^2 + \frac{\lambda(kl^2\lambda-1)}{2}\|\nabla\mathcal{L}_\theta(p^{s,a}, f_\theta^{s,a})\|^2$$
$$\leq -\frac{\lambda}{2}\|\nabla G(\theta_t)\|^2 + \frac{\lambda}{2}\|\nabla G(\theta_t) - \nabla\mathcal{L}_\theta(p^{s,a}, f_\theta^{s,a})\|^2$$

where the second equation is based on $\langle\mathbf{a}, -\mathbf{b}\rangle = \frac{1}{2}\left(\|\mathbf{a}-\mathbf{b}\|^2 - \|\mathbf{a}\|^2 - \|\mathbf{b}\|^2\right)$, and the last inequality is according to $\lambda \leq \frac{1}{kl^2}$. After taking the expectation, we have

$$\mathbb{E}\left[G(\theta_{t+1}) - G(\theta_t)\right]$$
$$\leq -\frac{\lambda}{2}\mathbb{E}\left[\|\nabla G(\theta_t)\|^2\right] + \frac{\lambda}{2}\mathbb{E}\left[\|\nabla G(\theta_t) - \nabla\mathcal{L}_\theta(p^{s,a}, f_\theta^{s,a})\|^2\right] \qquad (21)$$
$$\leq -\frac{\lambda}{2}\mathbb{E}\left[\|\nabla G(\theta_t)\|^2\right] + \frac{\lambda}{2}\left((1-\epsilon)^2\sigma^2 + \epsilon^2\kappa\sigma^2\right)$$

where the last inequality is based on Proposition 2. We take the summation, and therefore,

$$\mathbb{E}\left[G(\theta_T) - G(\theta_0)\right] \leq -\frac{\lambda}{2}\sum_{t=0}^{T-1}\mathbb{E}\left[\|\nabla G(\theta_t)\|^2\right] + \frac{T\lambda}{2}\left((1-\epsilon)^2\sigma^2 + \epsilon^2\kappa\sigma^2\right)$$

We let $\mathbb{E}\left[G(\theta_T)\right] = 0$ and $\epsilon = \frac{1}{1+\kappa}$, then,

$$\frac{1}{T}\sum_{t=0}^{T-1}\mathbb{E}\left[\|\nabla G(\theta_t)\|^2\right]$$
$$\leq \frac{2G(\theta_0)}{\lambda T} + (1-\epsilon)^2\sigma^2 + \epsilon^2\kappa\sigma^2$$
$$= \frac{2G(\theta_0)}{\lambda T} + \frac{2\kappa^2}{(1+\kappa)^2}\sigma^2 \qquad (22)$$
$$\leq \frac{2G(\theta_0)}{\lambda T} + 2\kappa^2\sigma^2$$

If $\kappa \leq \frac{\tau}{2\sigma}$ and let $T = \frac{4G(\theta_0)}{\lambda\tau^2}$, this leads to $\frac{1}{T}\sum_{t=0}^{T-1}\mathbb{E}\left[\|\nabla G(\theta_t)\|^2\right] \leq \tau^2$, i.e., $\tau$-stationary point, with the sample complexity as $O(\frac{1}{\tau^2})$. Thus, (2) has been proved. On the other hand, if $\kappa > \frac{\tau}{2\sigma}$, we set $T = \frac{G(\theta_0)}{\lambda\kappa^2\sigma^2}$. This implies that $\frac{1}{T}\sum_{t=0}^{T-1}\mathbb{E}\left[\|\nabla G(\theta_t)\|^2\right] \leq 4\kappa^2\sigma^2 = O(\kappa^2)$. Therefore, the degree of stationary point is determined the degree of distribution approximation measured by $\kappa$. Thus, we obtain (3). □

## F IMPLEMENTATION DETAILS

Our implementation is directly adapted from the source code in (Ma et al., 2020). For DAC (IQN), we consider the quantile regression for the distribution estimation on the critic loss. Instead of using fixed quantiles in QR-DQN (Dabney et al., 2018b), we leverage the quantile fraction generation

Table 1: Hyper-parameters Sheet.

| Hyperparameter | Value |
|---|---|
| *Shared* | |
|    Policy network learning rate | 3e-4 |
|    (Quantile / Categorical) Value network learning rate | 3e-4 |
|    Optimization | Adam |
|    Discount factor | 0.99 |
|    Target smoothing | 5e-3 |
|    Batch size | 256 |
|    Replay buffer size | 1e6 |
|    Minimum steps before training | 1e4 |
| *DAC (IQN)* | |
|    Number of quantile fractions ($N$) | 32 |
|    Quantile fraction embedding size | 64 |
|    Huber regression threshold | 1 |
| *DAC (C51)* | |
|    Number of Atoms ($k$) | 51 |

| Hyperparameter | $l_k$ for C51 | Max episode lenght |
|---|---|---|
| Walker2d-v2 | 500 | 1000 |
| Swimmer-v2 | 160 | 1000 |
| Reacher-v2 | 500 | 1000 |
| Ant-v2 | 500 | 1000 |
| HalfCheetah-v2 | 10,000 | 1000 |
| Humanoid-v2 | 5,000 | 1000 |
| HumanoidStandup-v2 | 15,000 | 1000 |
| BipedalWalkerHardcore-v2 | 50 | 2000 |

based on IQN (Dabney et al., 2018a) that uniformly samples quantile fractions in order to approximate the full quantile function. In particular, we fix the number of quantile fractions as $N$ and keep them in an ascending order. Besides, we adapt the sampling as $\tau_0 = 0, \tau_i = \epsilon_i / \sum_{i=0}^{N-1}$, where $\epsilon_i \in U[0, 1], i = 1, ..., N$.

### F.1 HYPER-PARAMETERS AND NETWORK STRUCTURE

We adopt the same hyper-parameters, which is listed in Table 1 and network structure as in the original distributional SAC paper (Ma et al., 2020).

### F.2 BEST $l_k$ FOR DAC (C51)

As suggested in Table 1, after a line search for the hyperparameter tuning, we select $l_k$ as 500, 10,000, 15,000, 160, 50, 5,000, 500, 500 for ant, halfcheetah, humanoidstand, swimmer, bipedalwalkerhardcore, humanoid, walker2d and reacher, respectively.

## G EXPERIMENTAL RESULTS ON ACCELERATION EFFECTS OF DISTRIBUTIONAL RL

**Same Architecture.** For a fair comparison, we keep the same DAC network architecture and evaluate the gradient norms of DAC (C51) and a variant of AC, which is optimized based on the expectation of the represented value distribution within the DAC implementation framework. Figure 4

suggests DAC (C51) still enjoys smaller gradient norms compared with AC in this fair comparison setting.

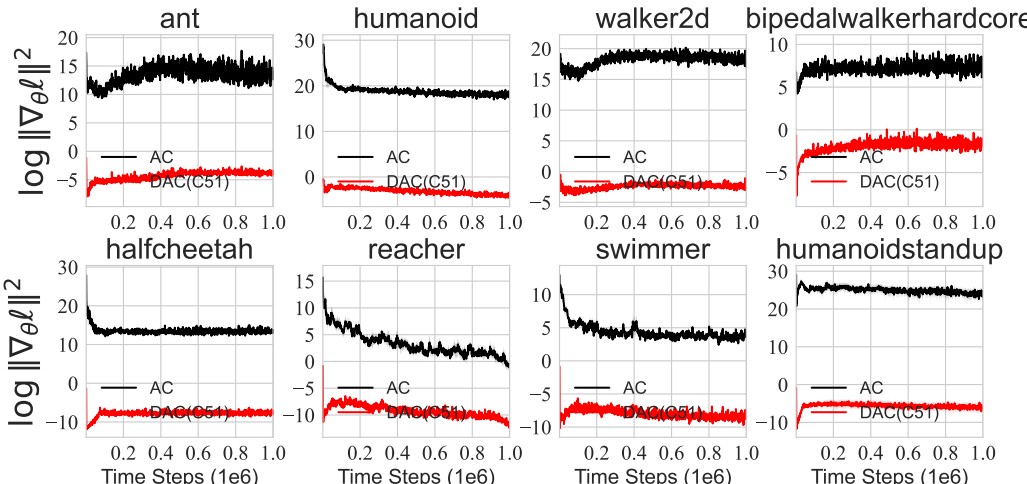

Figure 4: The critic gradient norms in the logarithmic scale during the training of AC and DAC (C51) over 5 seeds on three MuJoCo games. **We keep the same DAC network architecture and evaluate based on the expectation of the represented value distribution**.

**Results under Value Distribution Decomposition** We also provide gradient norms of both expectation and distribution based on the value distribution decomposition in Eq. 6. Similar results can be still observed in Figure 5.

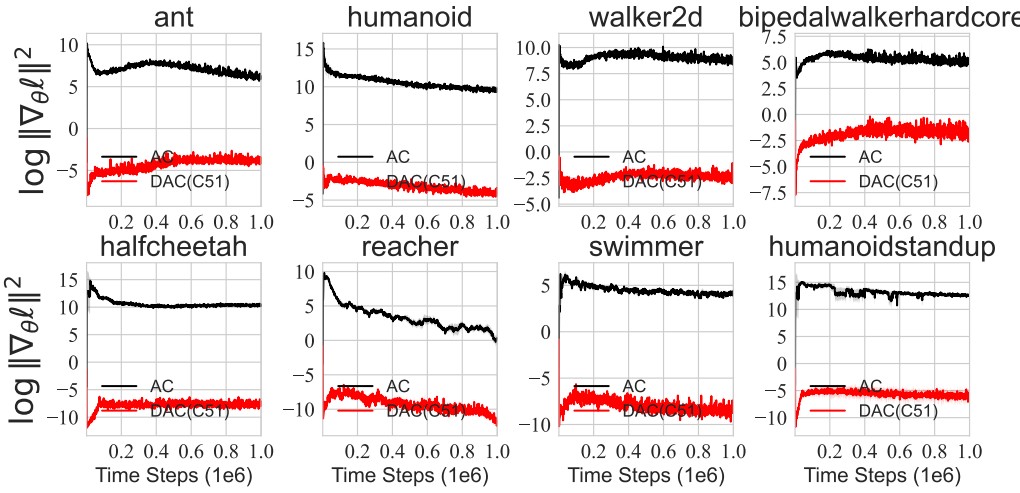

Figure 5: The critic gradient norms in the logarithmic scale during the training of AC and DAC (C51) over 5 seeds on three MuJoCo games. **Results of AC is the expectation part calculated via the Value Distribution Decomposition**.

# H  EQUIVALENCE BETWEEN THE LOSS FUNCTION IN THEOREM 2 AND MEAN SQUARED LOSS IN NEURAL FQI

**Proposition 3.** *(Connection between Theorem 2 and Mean Squared Loss in Neural FQI) In Eq. 2 of Neural FZI, if the function class $\{Z_\theta : \theta \in \Theta\}$ is sufficiently large such that it contains the*

*target* $\{Y_i\}_{i=1}^n$, *where* $Y_i = R(s_i, a_i) + \gamma Z_{\theta^*}^k (s_i', \pi_Z(s_i'))$. *Minimizing* $\mathcal{L}_\theta(\delta_{\{x=\mathbb{E}[Z^\pi(s,a)]\}}, f_\theta^{s,a})$ *in Theorem 2(1), as* $\max_i w_i \to 0$ *implies*

$$P(\widehat{Z}_\theta(s,a) = \mathcal{T}^{opt}Q_{\theta^*}(s,a)) = 1. \tag{23}$$

*Proof.* Bellman Optimality Operator is $\mathcal{T}^{opt}Q(s,a) = \mathbb{E}[R(s,a)] + \gamma \max_{a'} \mathbb{E}_{s' \sim p}[Q(s', a')]$. We also define the distributional Bellman optimality operator $\mathfrak{T}^{opt}$ as follows:

$$\begin{aligned}
\mathfrak{T}^{opt}Z(s,a) &\overset{D}{=} R(s,a) + \gamma Z(S', a^*) \\
S' &\sim P(\cdot \mid s, a), \quad a^* = \underset{a'}{\operatorname{argmax}} \mathbb{E}[Z(S', a')]
\end{aligned} \tag{24}$$

For the uniform notation, we ignore $k$ in Neural FZI. If $\{Z_\theta : \theta \in \Theta\}$ is sufficiently large enough such that it contains $\mathfrak{T}^{opt}Z_{\theta^*}$, then optimizing Neural FZI in Eq. 2 leads to $\widehat{Z}_\theta = \mathfrak{T}^{opt}Z_{\theta^*}$. Similarly, optimizing Neural FQI yields $\widehat{Q}_\theta = \mathfrak{T}^{opt}Q_{\theta^*}$ ideally.

We denote $w_E$ as the interval that $\mathbb{E}[Z^{\text{target}(s,a)}]$ and $Z^{\text{target}} = \mathfrak{T}^{opt}Z_{\theta^*}$ in expectation as shown in Neural FZI. $f^{s,a} \to q^{s,a}$ as $w_{\max} = \max_i w_i \to 0$, where $q^{s,a}$ is the continuous target probability density function. Then we have:

$$\begin{aligned}
&\mathcal{L}_\theta(\delta_{\{x=\mathbb{E}[Z^\pi(s,a)]\}}, f_\theta^{s,a}) \\
&= -\int_{x \in w_E} \delta_{\{x=\mathbb{E}[Z^{\text{target}}(s,a)]\}} \log f_\theta^{s,a}(w_E)dx \\
&\to -\log q_\theta^{s,a}(\mathbb{E}[Z^{\text{target}}(s,a)]) \quad \text{as} \quad w_E \to 0
\end{aligned} \tag{25}$$

where we know $\int \delta(x)dx = 1$. Since $\{Z_\theta : \theta \in \Theta\}$ is sufficiently large enough, the KL minimizer would be $\widehat{q}_\theta^{s,a} = \delta_{\mathbb{E}[Z^{\text{target}}(s,a)]}$, where $\delta_{\mathbb{E}[Z^{\text{target}}(s,a)]}$ is a Dirac Delta function centered at $\mathbb{E}[Z^{\text{target}}(s,a)]$ and can be viewed as a generalized probability density function. According to the definition of Dirac Delta function, as $w_{\max} \to 0$, we attain

$$\begin{aligned}
P(\widehat{Z}_\theta(s,a) &= \mathbb{E}[Z^{\text{target}}(s,a)] \\
&= \mathbb{E}[\mathfrak{T}^{opt}Z_{\theta^*}(s,a)]) \\
&= 1
\end{aligned} \tag{26}$$

Due to the linearity of expectation analyzed in Lemma 4 of (Bellemare et al., 2017a), we have

$$\begin{aligned}
\mathbb{E}[\mathfrak{T}^{opt}Z_{\theta^*}(s,a)] &= \mathfrak{T}^{opt}\mathbb{E}[Z_{\theta^*}(s,a)] \\
&= \mathcal{T}^{opt}Q_{\theta^*}(s,a)
\end{aligned} \tag{27}$$

Finally, for each iteration in Neural FZI, the following equation always holds:

$$P(\widehat{Z}_\theta(s,a) = \mathcal{T}^{opt}Q_{\theta^*}(s,a)) = 1 \quad \text{as} \quad \max_i w_i \to 0 \tag{28}$$

This indicates that the optimal value distribution $\widehat{Z}_\theta$ may take other values instead of the expectation, but the probabilities of these events happen are 0. This implies that minimizing the KL divergence in terms of the dirac Delta function is "almost" equivalent to the minimizer of mean squared loss. $\square$

