# OpenReview forum: "How Does Value Distribution in Distributional Reinforcement Learning Help Optimization?"
_ICLR.cc/2023/Conference — Submitted to ICLR 2023_

### Official Review · Reviewer_poww · 2022-10-24

**Confidence:** 4
**Correctness:** 2
**Technical Novelty And Significance:** 2
**Empirical Novelty And Significance:** 2
**Recommendation:** 3

**Clarity, Quality, Novelty And Reproducibility:**

=== Clarity ===

The work is written overall quite clearly. Presentation-wise, I would appreciate more paragraphs especially in Sec 3.3 where there is a big chunk of text that makes reading difficult. I would also appreciate in intro that the authors make more clear what their contributions are, in terms of characterizing the optimization effect of dist RL over value-based RL. Bullet points would be useful.

=== Quality ===

The paper's writing quality can be improved. The technical quality of the paper is mediocre because I think the theoretical components, though arguably should play a major role in the paper, are a bit underwhelming.

=== Novelty ===

Though the paper posits an arguably novel perspective in studying distributional RL's benefits over value-based RL, the arguments are not convincing enough to me. As a result, the work is not as novel in terms of contributions.

=== Reproduce ===

Overall experimental results should be reproducible given the source code.

**Details Of Ethics Concerns:**

No concerns.

**Strength And Weaknesses:**

=== Strength ===

The paper proposes an interesting conceptual question that may have been overlooked in prior literature, i.e., distributional RL may just help with optimizing the mean value by enjoying a better optimization landscape. While much of the conventional wisdom is that distributional RL helps with the side-effect of learning a better representation for value-based agents, this work posits a relatively new and orthogonal perspective.

=== Weakness ===

The theoretical results are a bit weak and cannot satisfactorily explain what happens in practice. I believe this gap renders the paper not very useful in obtaining insightful understanding of how and why distributional RL works better than value-based RL. I will expand more in the detailed comments.



**Summary Of The Paper:**

The paper proposes to understand the benefits of distributional RL through an optimization perspective. In fitted Z iteration and under categorical distributional parameterization, the paper analyzes the effect of distributional parameterization on the optimization problem, concretely a certain notion of stability (Thm 1) and guarantee to stationary point under finite sample (Thm 2). They also show some experiments to support the claim and showcase improvements of distributional RL over value-based RL in the control case.

**Summary Of The Review:**

A few questions.

=== **Goal of the paper and theory vs. practice** ===

A high level concern I have for the paper, is that the proposed theoretical explanation to justify the benefits of C51 over value-based RL, is a bit detached from practice. As a concrete example, Thm 1 shows that softmax parameterization entails certain uniform stability for the optimization problem. The logic here is that as a result of the stability, C51 should work better than just optimizing the expected value. This is not a satisfying explanation.

To see why, consider an algorithm that "represents values with a categorical distribution", but is not doing distributional RL. This algorithm effectively just does value-based RL, but represents the mean value as a softmax distribution instead of a scalar. This technique is commonly applied in a number of deep RL agents, which have proved useful empirically [1,2]. Note that such an algorithm does not use distributional back-up, so does not learn a proper distribution.

In other words, the arguments in Thm 1 explain why softmax parameterization may work better from an optimization perspective, but does not explain why learning a distribution is useful. However, learning distributions lies at the core of distributional RL, and failing to provide an explanation for the benefits of actually "learning a distribution" defeats the purpose of the paper.

[1] MuZero: Mastering Go, chess, shogi and Atari without rules, Schrittwieser et al, 2019
[2] Muesli: Combining Improvements in Policy Optimization, Hessel et al, 2020

Empirically, it is worth trying out the above algorithm to see if its performance is in between value-based RL and full C51. If it is indeed the case, then Thm 1 provides a partial explanation why the value-based categorical parameterized algorithm works better than value-based RL, but there is still a gap from full distributional RL or C51.

=== **Results in Thm 2** ===

Results in Thm 2 suffer from the same issue as Thm 1 -- the characterization of the stationary point comes purely from the softmax parameterization, but not from the fact that dist RL actually learns an approximation to the target return distribution. All the characterizations in Thm 2 and Thm 1 are based on softmax parameterization's advantage over scalar parameterization, but not why learning a dist is better than just learning the mean.

=== **Title and scope of the work** ===

Given that only softmax parameterization is analyzed in the work, I think it is better to modify the title of the paper as reflecting this. Quantile representations and QR-DQN type of algorithms are not included in the analysis.

=== **Experiments** ===

In the experiment, the authors have compared IQN variants of distributional actor-critic with other baselines, and have showed improvements. This is a bit confusing, because IQN uses quantile representations which are drastically different from C51 in terms of optimization landscape. Do we expect quantile representations to enjoy similar uniform stability and stationary point guarantee as C51? There is no proof in the paper. If not, where should we expect the performance improvements come from?

---

> ### Author Response · Authors · 2022-11-18
> **Author Response**
>
> Thank you for your valuable comments and suggestions. Below are our responses to your concerns. Please be free to let us know if there are any further questions.
>
> ### Goal of the paper and theory vs practice
> To the best of our understanding, the so-called distribution in [1,2] is used to **measure two policy distribution differences** in the policy optimization setting, while distributional RL is focused on value distribution rather than policy distribution. In addition, our categorical distributional loss is a **direct objective function** compared with mean squared loss in expectation-based RL. Thus, we argue that our analysis can directly answer the question of why learning a distribution is useful than not, although we use a categorical parameterization that may be in common with other techniques.
>
> Also, we agree that it would be more convincing to provide the in-between results of a ‘fake’ C51 that represents the distribution, but only optimizes based on their expectation. We will take this suggestion into account in the future.
>
> ### Results in Theorem 2
> Softmax parameterization is a typical strategy to represent the value distribution. Note that in order to investigate how the value distribution is useful **in general**, it is necessary to **dive deeper into a typical parameterization way**, e.g., softmax parameterization. Although it is neither perfect nor general, it is necessary and serves as the first step to more conclusions.
>
> ### Title and Scope of The work
> Our ambitious goal is to heuristically extend our conclusion to quantile-based distributional RL by providing more experiments. Narrowing down the title seems to be a good suggestion to match better with our analysis, although it may limit our conclusion to some extent.
>
> ### Experiments
> It is fair to have such a concern. Our ambitious goal is to heuristically extend our conclusion to quantile-based distributional RL by providing more experiments. The generality results in the fact that **representing value distribution might not be highly related to the way to represent it**. Thus, conclusions based on one typical parameterization, i.e., C51, are more likely to apply to other value representation ways, e.g., QRDQN and IQN.

---

> ### Comment · Area_Chair_6sNL · 2022-11-21
> **Any comments to the responses from the authors?**
>
> Dear Reviewer poww,
>
> Thank you very much for your insightful review.  The authors have provided responses to your concerns.  How did they change your evaluation?

---

### Official Review · Reviewer_bwzS · 2022-10-25

**Confidence:** 3
**Correctness:** 3
**Technical Novelty And Significance:** 2
**Empirical Novelty And Significance:** 2
**Recommendation:** 5

**Clarity, Quality, Novelty And Reproducibility:**

Much of my comments regarding clarity, quality, and novelty can be found in the previous discussion of strengths and weaknesses. The supplementary materials provide reasonably good detail on the hyperparameters and evaluation protocol used in the experiments.

**Strength And Weaknesses:**

### Strengths
- The paper combines a set of results from fairly disparate regions of the literature into a unified story to explain the benefits of distributional RL.

- The strategy of decomposing the distributional RL objective based on the mean value of the state and a perturbation distribution is appealing.

- The figures in section 4 are easy to read and interpret.


### Weaknesses

- Section 3.2:
  - The proof of smoothness of cross-entropy/KL is widely known and similar analysis is already provided by Imani and White.
  - The stability result is a straightforward corollary of the previous observations on stability of SGD and smoothness of the CE loss. These bounds are also not very informative of generalization in practice because they are vacuous for optimization trajectories that don’t converge to a low loss in less than one epoch.

- Section 3.3:
  - The notation in section 3.3 is unclear and makes it difficult to follow precisely what the claim is that is stated in Proposition 2.
  - The notation $G^k(\theta)$ is not obvious: what is the interpretation of $k$? Is it the loss with respect to the current targets at iteration $k$ of FZI? Why not include this dependence elsewhere in that case?
    - Is f_\theta^{s,a} meant to be the density of the true return, or of the bellman targets?
    - The overloading of $\mathcal{L}_\theta$ is ambiguous as the paper does not specify which argument corresponds to the prediction and which to the target distribution, and hence which order these appear in the KL divergence.
	-I am assuming that the notation $\nabla G$ corresponds to the expectation of the left hand side gradient in equations 7 and 8 from context, but it is defined as the expectation of $L(s,a)$ in the paragraph previously, which was previously defined to be shorthand for the gradient of the current function approximator predictions with respect to the true FZI targets. This inconsistency makes it difficult to interpret the proposition 2 and theorem 2.
	- The result on distributional regularization would benefit from clarification on the role of $\kappa$ and $\sigma^2$. In the case of a deterministic MDP it seems that case (2) should be identical to case (1), so I assume that instead the assumptions required by theorem 2 preclude this type of prediction target, in particular due to the lower bound on the value of $\kappa$. In this case it seems that the assumptions of theorem 2 are relatively restrictive — what amount of noise in practice would be needed to fall into this sweet spot where convergence is faster than deterministic targets but still guarantees a $\tau$-stationary point?

- Section 4

- The use of actor-critic algorithms in section 4 is not well-motivated. Prior work (Ilyas et al.) has shown that the critic value estimates in these architectures are often wildly inaccurate and yet still provide significant benefits to performance. Thus it cannot be concluded that any performance improvement obtained by altering the critic architecture is caused by improved value prediction accuracy or stability of the critic.
- The IQN baseline isn’t informative as it follows a markedly different training procedure from that studied in previous sections. I would prefer to have seen a QR-DQN baseline with a similar parameterization as the C51 agent for a more reasonable comparison.
- The empirical results exhibit a similar ranking as is predicted by the theoretical results, though the convergence rates don’t seem to be consistent.
- Section 3 focuses on a single iteration of FZI whereas SAC follows an entirely different learning algorithm. It is not clear to me why we should expect the findings on convergence to a fixed set of targets to hold for the SAC algorithm, which updates the targets more frequently.


**Summary Of The Paper:**

This paper analyzes the optimization dynamics induced by distributional losses. It does so using two main classes of theoretical tools: first, it studies the smoothness of distributional losses and the consequent stability of gradient descent on these losses. Second, it studies the variance of the gradients obtained by distributional losses and characterizes the convergence rate of gradient descent. The qualitative predictions of the theoretical results are evaluated on two distributional algorithms.

**Summary Of The Review:**

This paper addresses an interesting problem, but suffers from a lack of clarity and technical depth. Of its main contributions, the result on stability and smoothness of distributional RL losses are straightforward corollaries of previously known results. The analysis of the convergence rate of distributional vs expected updates is not clear, and it is difficult to interpret how restrictive the necessary assumptions are for Theorem 2 (in particular case (2)) to remain valid. Finally, there is a fairly large gap between the setting of the theoretical analysis and the empirical experiments, which limits the insight these experiments are able to provide.

---

> ### Author Response · Authors · 2022-11-18
> **Author Response**
>
> Thank you for your valuable comments and suggestions. Below are our responses to your concerns. Please be free to let us know if there are any further questions.
>
> ### Section 3.2
> * Imani and White mainly provided a similar proof of Lipschitz continuity, while our Proposition 1 extends their analysis by additionally providing the **Lipschitz smoothness**, i.e., the Lipschitz continuity on the gradients. Note that Lipschitz smoothness is also crucial to derive the uniform stability in Theorem 1, which consists of our contribution.
> * Previous observations on the stability of SGD normally assume smoothness assumptions of a general loss function, while we specifically analyze the stability of optimizing categorical distribution loss based on the observations of the smoothness property of categorical distributional loss. All of these analyses are also applicable based on Neural FZI framework we establish. We are not sure about your comment about the useless generalization bound in practice, and it would be appreciated if you could provide some references.
>
> ### Section 3.3
> * Based on our knowledge of SGD convergence, a typical strategy is to measure the variance of gradient estimates, which plays a key role in the convergence analysis. Based on this principle, Proposition 2 is a direct corollary of Eq. 8, which is used to derive Theorem 2 eventually.
> * $k$ in $G^k$ represents the k-iteration in Neural FZI and we ignore it for simplicity in Section 3.3.
>   * $f_\theta^{s, a}$ is the parameterized density function.
>   * We apologize for the definition confusion of $G_\theta$. To address it, we revised our clarity and pointed out $G^k(\theta) = \mathbb{E}\left[\mathcal{L}_\theta(\delta(\{x=\mathbb{E}\left[Z^\pi(s, a)\right]\}), f_\theta^{s, a})\right]$, **which is the expectation of loss function in terms of the Dirac Delta function**.
>   * Maybe we did not understand this question, but we are trying to explain. When the parameterized value distribution can approximate the true value distribution favorably, the variance of gradient estimates measured by $\kappa$ is more likely to be small than $\frac{\tau}{2\sigma^2}$. In this case, we have a faster convergence (smaller iteration complexity) as well as a $\tau$-stationary point guarantee according to Theorem 2 (2).
> .
> ### Section 4
> * Ilyas et al.’s work in fact reflects there is still a gap between the theoretical analysis and algorithm performance in practice. Although there might be some gap, our stability analysis is in line with our empirical results, which can at least provide partial evidence to demonstrate our conclusion to the best of the whole research community’s knowledge.
> * Our current version has not involved such comparison and we will take this suggestion into account in the revised version in the future. This may involve heavy work as conducting quantile regression based on categorical distribution parameterization is not straightforward.
> * The convergence rate in the tabular setting can partially account for the whole convergence speed as we also need to consider the optimization factor as analyzed in Section 3.1. In other works, as stated in Section 3.1, the whole optimization speed not only depends on the contraction rate of Bellman update related to a particular distribution divergence, but also is linked with the convergence speed to solve the optimization problem within each iteration in Neural FZI.
> * We would like to clarify that **SAC also applies a target network strategy** that perfectly fits into Neural FZI framework as well.

---

> > ### Comment · Reviewer_bwzS · 2022-11-27
> > **Thanks for the clarifications**
> >
> > Thanks to the authors for the clarifying answers to my questions, particularly on Section 3.3. To address the question about the usefulness of the generalization bound, my concern about its vacuousness is a direct result of the form of the bound rather than stemming from follow-up published work. Specifically, the T/n term will usually need to be less than 1 for the resulting generalization bound to be non-vacuous.

---

> ### Comment · Area_Chair_6sNL · 2022-11-21
> **Any comments to the responses from authors?**
>
> Dear Reviewer bwzS,
>
> Thank you very much for your detailed review.  Did the responses from the authors clarify what was unclear?  Did the clarification change your evaluation on the technical depth?

---

### Official Review · Reviewer_d87u · 2022-10-25

**Confidence:** 3
**Correctness:** 2
**Technical Novelty And Significance:** 3
**Empirical Novelty And Significance:** 1
**Recommendation:** 3

**Clarity, Quality, Novelty And Reproducibility:**

The paper is generally well-written and the setup is described with great clarity. I have a small concern for discussion text below Proposition 1 on Page 5 -- I found the the argument concerning the stability of the classical RL compared to distributional RL to be a bit loose. The authors argue that the mean square error in classical RL can lead to large gradients because they scale with the Q-value which can be potentially large, whereas the distributional RL gradient scale with $k$ (which is the number of bins in the distributional RL) which can be much smaller than the Q-value. However, distributional RL can also require a large $k$ to obtain the desired precision and it is unclear how these two values (gradient norm of neural FQI vs. neural FZI) can be compared without grounding in a more concrete example/setup. In fact, higher $k$ should in general lead to lower approximation error of the distribution (e.g., $\kappa$ in the analysis later in the paper), which can greatly impact the convergence rate of neural FZI.

Some other minor text errors and missing references:
- Page 2, "a recent progress" => "recent progress"
- Page 2, "has also be revealed" => "has also been revealed". Also the sentence does not read well.
- Page 2, "Empirical results collaborate that distributional RL indeed
enjoys a stable gradient behavior..." --  "collaborate" => "corroborate"?
- Page 3, "Neural FQI is exactly the updating under ..." -- "updating" => "update"?
- Page 4, "distributional Bellman operator under Cramer distance is $\gamma^{1/2}$-contractive and is a $\gamma$-contraction when $d_p$ is Wasserstein distance" -- missing a reference here?

**Strength And Weaknesses:**

*Strength*

- The theoretical results in the paper look reasonable and likely to be correct. Though admittedly I did not check all the proof details in the appendix.
- The paper is doing a great job of positioning itself in the literature.

*Weaknesses*

- The paper argues that there are acceleration effects of distributional RL compared to classical RL which optimizes the mean square error $(y\_i - Q\_\theta^k(s_i, a_i))^2$. However, this is not well supported by the theorem as the main result that demonstrates the acceleration effects do not use the mean square error. Rather it uses KL with the target density being the dirac delta of expectation of the value target (see Theorem 2 (1) -- $\mathcal{L}\_\theta(\delta\_{x=\mathbb{E}\left[Z^\pi(s, a)\right]}, f\_\theta^{s, a})$). It is not obvious to me how the results under this KL objective may be translated to the mean square error objective.

- The empirical results do not support the theory well. While the paper shows that the critic gradient norm with respect to the parameters for distributional RL is smaller, this comparison is confounded by the difference in the objective (e.g., AC uses mean square error, C51 uses KL, and IQN uses Wasserstein distance).

**Summary Of The Paper:**

The paper studies the convergence rate of neural fitted Z-iteration with a discrete distribution approximation of the Q distribution. In particular, the authors demonstrate that under the linear categorical parameterization assumption, neural fitted Z-iteration converges faster with the distributional RL objective compared to the classical RL objective (where the expectation is used as the fitting target rather than a distribution). The paper also shows that the gradient norm of the critics with respect to the network parameters for practical distributional RL algorithms is generally lower than the gradient norm for practical non-distributional RL algorithms, corroborating the theoretical results.

**Summary Of The Review:**

While the theoretical contributions are interesting on their own, my biggest concern of the paper at its current state is that both the theoretical results and the empirical results do not demonstrate why distributional RL is more stable than classical RL (which from my understanding is the main point of the paper). Specifically, the theoretical results do not analyze the correct objective for the classical RL (see above) and because of it, the empirical results that aim to corroborate the theoretical insights are also confounded by the objective mismatch. Therefore, I would not recommend acceptance of the paper.

---

> ### Author Response · Authors · 2022-11-18
> **Author Response**
>
> Thank you for your valuable comments and suggestions. Below are our responses to your concerns. Please be free to let us know if there are any further questions.
>
> ## Weakness 1: Equivalence between mean squared loss with KL regarding a Dirac delta function
> Thank you for raising this insightful question. To address this issue, we provide a new **Proposition 3 of Appendix H** to demonstrate that the minimizer of the KL divergence in terms of the Dirac delta function is equivalent to the one of mean squared loss in Neural FQI as the interval length of histogram density estimator tends to 0. The detailed proof is provided in Appendix, and we hope this proposition could help to clarify your concern.
>
> ## Weakness 2. Confounded Loss functions
> Note that Mean squared error in AC, KL in C51 and Wasserstein distance in IQN all range in $[0,+\infty)$, and thus it ought to be fair to compare the gradient norms under these different loss functions. We also provide gradient norms under the same architecture in Figure 4 of Appendix G to rule out the impact of model architecture difference, and similar results are also observed.
>
> ## Clarity, Quality, Novelty and Reproducibility
> In C51, increasing $k$ would saturate the network capacity quickly and increases the computation overhead, which is not applicable in practice. Moreover, an overly large $k$ may lead to the overfitting issue or increase the training instability issue arising from the leverage of larger network capacity. A similar discussion is provided in [1].
>
> ## Minors.
> We thank you for these suggestions. We revised based on your suggestions.
>
> [1] Nils Bjorck, Carla P. Gomes, and Kilian Q. Weinberger. Towards deeper deep reinforcement learning with spectral normalization. (NeurIPS 2021)

---

> > ### Comment · Reviewer_d87u · 2022-11-26
> > **Reviewer Response**
> >
> > Thanks for your response!
> >
> > >  new Proposition 3 of Appendix H
> >
> > Thanks for adding the new results! It certainly helps to know that the minimizers of two loss functions are the same. However, such equivalence does not guarantee that the optimization processes of the two loss functions are similar. It is still not obvious to me how the stable optimization results (Theorem 2) can be translated into the mean square error objective since the existing results require analysis of the optimization process.
> >
> > > confounded loss functions
> >
> > Thanks for adding the additional results in Appendix G. Ruling out the architecture discrepancy definitely helps in making the comparison fairer. However, my main concern regarding the loss functions still remains (related to the discussion above). In particular, I am mainly concerned about the fact that the objective that the authors analyzed in the theory section is different from the objectives used in the experiment section. In particular, the authors analyzed the KL objective (e.g., in Theorem 2), whereas in the experiment section, three different objectives were used (MSE for AC, KL for C51, and W-distance for IQN). The comparison among these three different objectives cannot be used to support the theory because these are different from the one being analyzed. Hope this clarifies my concern!

---

> ### Comment · Area_Chair_6sNL · 2022-11-21
> **Any comments to the responses from authors?**
>
> Reviewer d87u,
>
> Thank you very much for your informative review.  The authors have provided responses to your concerns and revised the paper.  How did they change your evaluation?  Does new Proposition 3 resolve your concern on the theoretical results?

---

> > ### Comment · Reviewer_d87u · 2022-11-26
> > **My Evaluation Remains the Same**
> >
> > Dear AC,
> >
> > I have read the author response and I believe my main concern over the discrepancy in the loss functions (KL for distributional RL and MSE for classical RL) still remains. The authors provided a new theoretical result that shows the *minimizers* of the two loss functions in question (KL vs. MSE) are the same. However, this result does not imply that the main results (e.g., Theorem 2) can be easily modified from the KL objective to the MSE objective since the gradients of these objectives are not guaranteed to behave similarly. I would like to keep my current evaluation of the paper for this reason.

---

### Author Response · Authors · 2022-11-18
**General Author Response**

Dear Reviewers and ACs,

Thanks again for your useful comment and insightful suggestions. Based on your advice, we revised our paper accordingly (see the revised pdf) by providing new proof to show the equivalence in Theorem 2(1) and improved our clarifications.  We also provide the original paper in submission as well as the source code in the supplementary file for reference. It would be much appreciated if you can take the revision above into your final consideration for our paper. Please be free to let us know if you have any further questions.

Best Wishes,
Authors

---

### Decision · Program_Chairs · 2023-01-20

**Decision:**

Reject

**Justification For Why Not Higher Score:**

The main claim is not well supported either by theory or by experiments.

**Justification For Why Not Lower Score:**

N/A

**Metareview: Summary, Strengths And Weaknesses:**

This paper investigates why distributional reinforcement learning (RL) can outperform conventional expectation-based RL.  The paper provides theoretical and empirical support for this claim.  The theoretical results are interesting in its own right and give new insights into distributional RL, which constitute the major strength of the paper.

However, there are some major gaps between theory and claims, and the main claim is not well supported by the theory.  Experimental results do not well support the claim either.